# VIBRA: Redundancy-Aware Information Bottleneck for Hallucination-Resistant Vision-Language Models

## Abstract

Vision-Language Models (VLMs) have achieved impressive progress across a range of multimodal tasks but remain highly susceptible to visual hallucination, producing text that contradicts the visual input. Existing mitigation strategies often rely on additional large-scale VLMs or multi-stage decoding, which hinders efficiency and broad applicability. In this work, we identify redundant and noisy image features as a primary cause of hallucination, as they degrade the model's ability to capture semantically relevant visual content. Correspondingly, we propose VIBRA (Vision-Language Information Bottleneck with Redundancy Awareness), a plug-and-play module that adaptively filters out redundant visual information while preserving task-relevant semantics at both the token and feature levels. Specifically, VIBRA employs a multi-modal information bottleneck to retain image features aligned with textual input and introduces adaptive token filtering through spectral clustering and compression-aware pruning to eliminate instance-specific redundancy. Additionally, we design a Binary-Guided loss to sharpen the separation between informative and noisy features, enabling more effective visual information gating. Extensive experiments demonstrate that VIBRA consistently enhances visual reasoning and reduces hallucination across a variety of VLM architectures.

## 1 Introduction

Vision-Language Models (VLMs) have achieved remarkable progress across a variety of multimodal tasks, including image captioning, visual question answering (VQA), and multimodal reasoning (Liu et al., 2023b; Zhang et al., 2023b; Zhu et al., 2023; Liu et al., 2024c). These models have demonstrated broad applicability in diverse domains, including healthcare, education, and human–computer interaction. Despite their impressive visual understanding capabilities, however, VLMs remain prone to visual hallucination, generating textual outputs inconsistent with the visual input (Biten et al., 2022; Gunjal et al., 2024; Li et al., 2023b). Such hallucinations compromise reliability and applicability, particularly in safety-critical settings (Liu et al., 2023a; Lovenia et al., 2023; Dai et al., 2022; Guan et al., 2024).

To mitigate hallucinations, prior work has focused on self-refinement (Zhou et al., 2023; Huang et al., 2024; Yin et al., 2024) and decoding strategies (Chuang et al., 2023; Chen et al., 2024b; Leng et al., 2024). However, self-refinement often depends on additional large-scale VLMs, while decoding-based approaches typically involve multi-round generation and rollback, which significantly increases inference latency and limits practicality. Other approaches improve modality alignment via lightweight projectors or Q-Formers (Li et al., 2023a; Dai et al., 2023; Gao et al., 2023; Chen et al., 2023; Liu et al., 2024b), but still process all image tokens, including redundant ones.

Recent studies (Shang et al., 2024; Chen et al., 2024a; Zhang et al., 2025; Gong et al., 2024) suggest that not all image tokens contribute equally to VLM reasoning. In fact, selectively removing redundant or irrelevant image tokens can improve performance and faithfulness. As shown in Figure 1, using fewer, more semantically relevant tokens often outperforms indiscriminate use of all visual tokens.

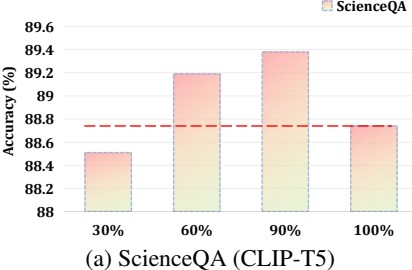 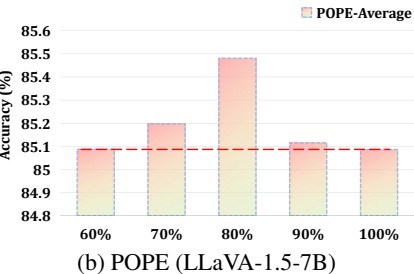

(a) ScienceQA (CLIP-T5)          (b) POPE (LLaVA-1.5-7B)

Figure 1: Effect of retaining only the top-$k\%$ image tokens ranked by image–text cosine similarity. Figure (a) evaluates the CLIP-T5 multimodal model on the ScienceQA dataset, while figure (b) tests LLaVA-1.5-7B on the POPE hallucination-detection benchmark. Across both datasets and models, varying the proportion of retained image tokens consistently demonstrates that a small subset of semantically relevant image tokens can outperform the full-token baseline, including improvements in reasoning accuracy and reductions in hallucination-related errors.

Several methods attempt to leverage this insight. FastV (Chen et al., 2024a) and LLaVA-PruMerge (Shang et al., 2024) prune tokens based on attention weights, but risk discarding semantically relevant content due to attention's unreliability (Gong et al., 2024; Darcet et al., 2023). Other methods, such as Simignore (Zhang et al., 2025) and PuMer (Cao et al., 2023), retain top-k image tokens ranked by image-text similarity but use fixed thresholds, ignoring instance-specific variation in visual complexity. These rigid filtering schemes limit adaptability and effectiveness. Moreover, selecting informative image tokens solely based on image-text cosine similarity, as adopted in many existing works, often fails to identify the most semantically meaningful tokens. To address this, we leverage the Multi-Modal Information Bottleneck (MIB) Compression Term, which quantifies the information contribution of each image token, providing a more effective criterion than the commonly used cosine similarity strategy for selecting semantically informative tokens (see section 5.3 for details).

We argue that effective visual understanding in VLMs requires adaptive filtering of semantically irrelevant image tokens, along with fine-grained suppression of irrelevant visual information at the feature level. Motivated by these insights, we propose VIBRA (Vision-Language Information Bottleneck with Redundancy Awareness), a plug-and-play module designed to suppress visual redundancy at both the token and feature levels (Schulz et al., 2020). VIBRA restricts irrelevant visual signals by injecting noise into image features and learning an information bottleneck parameter guided by textual semantics. This mechanism enables the model to effectively "turn off" text-irrelevant visual signals while retaining semantically meaningful content, thereby suppressing feature-level visual redundancy. To further support instance-specific token pruning, we introduce Adaptive Image Token Filtering via Spectral Clustering, which removes redundant visual tokens via spectral clustering and compression-aware pruning. Additionally, a Binary-Guided loss is proposed to supervise the bottleneck, encouraging it to make near-binary decisions between informative content and noisy signals.

We demonstrate the effectiveness of VIBRA on two representative tasks: image reasoning and faithful visual understanding. When integrated in a plug-and-play fashion with representative Multimodal Reasoning Models (MM-COT$_{T5-Base}$ (Zhang et al., 2023b), MC-COT$_{T5-Base}$ (Tan et al., 2024)) and Large Vision-Language Models (MiniGPT-4 (Zhu et al., 2024), LLaVA-1.5 (Liu et al., 2024b)), VIBRA achieves consistent improvements without additional inference stages. Our main contributions are as follows:

• We propose VIBRA, a novel framework that employs a redundancy-aware information bottleneck to effectively suppress irrelevant visual signals while preserving semantically meaningful content at both token and feature levels.

• We introduce three core components that work synergistically within VIBRA: a Variational Multi-Modal Information Bottleneck that regulates visual information flow based on text semantics, an Adaptive Image Token Filtering mechanism using Spectral Clustering and compression-aware pruning to eliminate image-specific redundancies, and a Binary-Guided loss that enforces a sharp distinction between informative and noisy features for improved visual reasoning.

• Through extensive experiments, we show that VIBRA consistently improves performance and reduces visual hallucination across multiple VLMs and benchmarks.

## 2 RELATED WORK

**VLMs for Visual Understanding and Reasoning.** VLMs have advanced a wide range of tasks such as image captioning, VQA, and multimodal reasoning. Foundational models like CLIP (Radford et al., 2021) and ALIGN (Jia et al., 2021) align visual and textual modalities via contrastive learning, but are mainly suited for retrieval and recognition rather than generation or open-ended reasoning. To bridge this gap, recent large VLMs (Liu et al., 2023b; 2024b; Dai et al., 2023; Bai et al., 2023; Chen et al., 2024c; Ye et al., 2024; Zhu et al., 2023) integrate visual signals into pre-trained large language models (LLMs) via cross-modal projection modules and instruction tuning, significantly enhancing multimodal generation and reasoning capabilities.

A growing body of work extends these models to multimodal reasoning, which demands not only accurate answers but also interpretable rationales. Inspired by chain-of-thought (CoT) prompting in LLMs, MM-CoT (Zhang et al., 2023b) introduced a two-stage reasoning framework: generating a rationale from an image-question pair, then predicting the answer based on that rationale. Subsequent approaches (Mondal et al., 2024; He et al., 2024; Tan et al., 2024; Wang et al., 2024a) have refined this paradigm, achieving stronger reasoning performance in open-ended settings.

**Reducing Visual Hallucination.** Visual hallucination, where models generate descriptions misaligned with the actual visual content, remains a key challenge that undermines the reliability of VLMs (Rohrbach et al., 2018; Guan et al., 2024). To mitigate visual hallucination, a variety of strategies have been explored, including post-hoc correction (Zhou et al., 2023; Huang et al., 2024; Yin et al., 2024; Feng et al., 2024; Wu et al., 2024), alternative decoding methods (Chuang et al., 2023; Chen et al., 2024b; Leng et al., 2024; Kim et al., 2024; Wang et al., 2024b; Zhu et al., 2025), and instruction tuning on meticulously curated datasets (Yue et al., 2024; Ma et al., 2024; Yu et al., 2024; Liu et al., 2024a; Gunjal et al., 2024).

However, these approaches present significant limitations. Instruction tuning typically demands large volumes of high-quality annotated data, which are costly and labor-intensive to obtain. Post-hoc refinement methods often rely on auxiliary large-scale VLMs, introducing substantial inference overhead. Similarly, decoding-based solutions commonly involve multiple rounds of generation and rollback, leading to increased latency and computational expense. In this paper, we propose a lightweight, plug-and-play module that mitigates visual hallucination by strategically filtering redundant visual information. Our approach improves factual alignment in VLM outputs while avoiding the inference inefficiencies and data burdens associated with prior methods.

**Information Bottleneck.** The Information Bottleneck (IB) (Tishby et al., 2000) principle provides a theoretical foundation for learning representations that balance compression of input data with preservation of task-relevant information. It has been widely applied across domains including language modeling (Li & Eisner, 2019; West et al., 2019), subspace explaining (Miao et al., 2022; Liu et al., 2024d), and deep model interpretability (Schulz et al., 2020; Wang et al., 2023).

Formally, the IB objective aims to learn a latent representation $\mathbf{Z}$ from input $\mathbf{X}$ that retains maximal relevant information about a target variable $\mathbf{Y}$, while minimizing irrelevant information from $\mathbf{X}$:

$$\min \mathcal{L} = \underbrace{\beta \cdot I(\mathbf{X}; \mathbf{Z})}_{\text{Compression Term}} - \underbrace{I(\mathbf{Z}; \mathbf{Y})}_{\text{Supervision Term}} , \tag{1}$$

where $I(\,\cdot\,;\,\cdot\,)$ denotes mutual information and $\beta$ controls the trade-off between compression and informativeness.

Unlike most prior work focused on single-modality or self-supervised learning, we incorporate cross-modal supervision by using textual features to guide the visual bottleneck. This allows VIBRA to preserve semantically aligned image features while suppressing task-irrelevant visual redundancy, leading to improved performance in vision-language reasoning.

## 3 PRELIMINARIES

**Vision-Language Model Setup.** A typical Vision-Language Model (VLM), denoted $\theta$, consists of three core components: unimodal encoders, a cross-modal alignment module, and a language

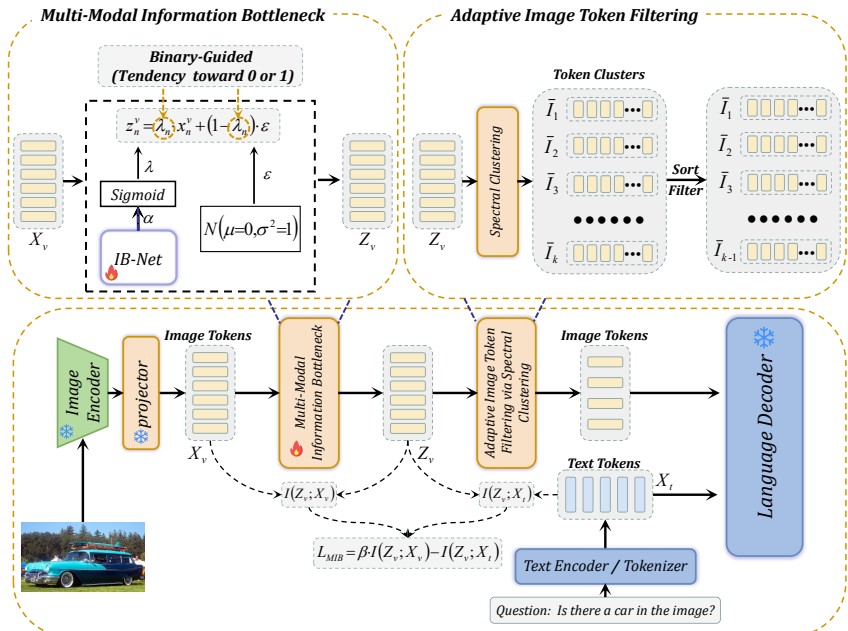

Figure 2: The illustration of the proposed VIBRA framework, which consists of (1) Variational Multi-Modal Information Bottleneck; (2) Binary-Guided Token Importance Modeling; and (3) Adaptive Image Token Filtering via Spectral Clustering.

model decoder. These components jointly transform multimodal inputs into coherent, grounded textual outputs.

Specifically, the textual input is first tokenized into a sequence of text tokens $\mathbf{X}_t$ using a tokenizer or a pretrained text encoder (e.g., T5 Encoder (Raffel et al., 2020)). Concurrently, the image input is processed by a visual encoder (e.g., CLIP (Radford et al., 2021)) to produce a sequence of visual tokens. These tokens are then passed through a cross-modal projection or alignment layer, which transforms them into modality-aligned visual representations $\mathbf{X}_v$ that are compatible with the language model's input space.

The language decoder receives both $\mathbf{X}_t$ and $\mathbf{X}_v$ and autoregressively generates the output sequence $y = \{y_1, y_2, \ldots, y_T\}$. During training, the model is optimized using the standard cross-entropy loss:

$$\mathcal{L}_{CE} = \mathbb{E}\left[-\log\left(p_\theta(y_t \mid y_{<t}, \mathbf{X}_v, \mathbf{X}_t)\right)\right], \tag{2}$$

where $p_\theta$ denotes the probability of generating token $y_t$ conditioned on the previously generated tokens $y_{<t}$, as well as the visual and textual inputs. This objective encourages the model to learn accurate, context-aware generation grounded in both modalities.

## 4 THE PROPOSED METHOD

### 4.1 OVERVIEW

We propose VIBRA (Vision-Language Information Bottleneck with Redundancy Awareness) to suppress redundant visual signals at both the *feature-level* and *token-level*, thereby enhancing image understanding in vision-language models. The core idea leverages a multimodal information bottleneck mechanism to filter irrelevant visual content, guided by compression-aware metrics for token filtering. Specifically, VIBRA consists of three key components: (1) **Variational Multi-Modal Information Bottleneck**, (2) **Binary-Guided Token Importance Modeling**, and (3) **Adaptive Image Token Filtering via Spectral Clustering**. The overall architecture is illustrated in Figure 2.

## 4.2 Variational Multi-Modal Information Bottleneck

To preserve semantic information at the feature level, we introduce a Multi-Modal Information Bottleneck (MIB) module following the frozen pre-trained image encoder, as shown in Figure 2. Let $\mathbf{X}_v = [x_1^v, \ldots, x_N^v] \in \mathbb{R}^{d \times N}$ and $\mathbf{X}_t = [x_1^t, \ldots, x_M^t] \in \mathbb{R}^{d \times M}$ denote the visual and textual feature representations extracted by the image and text encoders, respectively, where $N$ and $M$ denote the number of image and text tokens, and $d$ is the feature dimension.

For *feature-level* compression, we inject controlled noise into the visual features using a learnable gating mechanism. The latent representation $\mathbf{Z}_v = [z_1^v, \ldots, z_N^v] \in \mathbb{R}^{d \times N}$ is defined as:

$$z_n^v = \sigma(f_{\text{IB}}(x_n^v)) \cdot x_n^v + (1 - \sigma(f_{\text{IB}}(x_n^v))) \cdot \varepsilon, \tag{3}$$

where $\varepsilon \sim \mathcal{N}(0,1)$ is Gaussian noise, and $\sigma(\cdot)$ denotes the sigmoid function. The function $f_{\text{IB}}$ represents IB-Net, which consists of four linear layers with ReLU activations between them and a final sigmoid activation that outputs a gating scalar $\lambda_n \in [0,1]$ for each token. This formulation enables feature-level information compression: when $\lambda_n$ is close to 0, the original feature $x_n^v$ is largely replaced by noise, effectively suppressing redundant information. Conversely, when $\lambda_n$ is close to 1, the feature is preserved.

Our goal is to ensure that the retained image tokens capture rich semantic information from the corresponding text. To this end, we introduce the mutual information term $I(\mathbf{Z}_v; \mathbf{X}_t)$ to quantify the amount of semantic information from the textual modality is encoded in the compressed visual representation $\mathbf{Z}_v$. Meanwhile, $I(\mathbf{Z}_v; \mathbf{X}_v)$ measures how much visual information is retained from the original image features.

Based on Eq. 1, we design an objective to optimize the parameters of IB-Net as follows:

$$\mathcal{L}_{\text{MIB}} = \underbrace{\beta \cdot I(\mathbf{Z}_v; \mathbf{X}_v)}_{\text{Compression Term}} - \underbrace{I(\mathbf{Z}_v; \mathbf{X}_t)}_{\text{Supervision Term}}, \tag{4}$$

where $\beta$ is a hyperparameter that balances the trade-off between the compression term and the supervision term.

Since mutual information cannot be computed exactly in general, it is often estimated through variational approximation. Following (Alemi et al., 2016; Schulz et al., 2020), the upper bound of $I(\mathbf{Z}_v; \mathbf{X}_v)$ can be computed as:

$$\begin{aligned} I(\mathbf{Z}_v; \mathbf{X}_v) &= \mathbb{E}_{\mathbf{X}_v} \left[ D_{\text{KL}} \left( P(z_v \mid x_v) \,\|\, P(z_v) \right) \right] \\ &= \mathbb{E}_{\mathbf{X}_v} \left[ D_{\text{KL}} \left( P(z_v \mid x_v) \,\|\, Q(z_v) \right) \right] - D_{\text{KL}} \left( P(z_v) \,\|\, Q(z_v) \right) \\ &\leq \mathbb{E}_{\mathbf{X}_v} \left[ D_{\text{KL}} \left( P(z_v \mid x_v) \,\|\, Q(z_v) \right) \right], \end{aligned} \tag{5}$$

where $D_{\text{KL}}(\cdot \,\|\, \cdot)$ denotes the Kullback-Leibler divergence.

According to Eq. 3, the conditional distribution of the latent representation $z_n^v$ given the visual token $x_n^v$ is defined as: $z_n^v \mid x_n^v \sim \mathcal{N}(\mu = \lambda_n x_n^v, \sigma^2 = (1 - \lambda_n)^2)$, while the marginal distribution is assumed to be: $z_n^v \sim \mathcal{N}(\mu = 0, \sigma^2 = 1)$. Consequently, $I(\mathbf{Z}_v; \mathbf{X}_v)$ can be upper-bounded by the KL divergence between these two Gaussian distributions:

$$\begin{aligned} &\mathbb{E}_{\mathbf{X}_v} \left[ D_{\text{KL}} \left( P(z_v \mid x_v) \,\|\, Q(z_v) \right) \right] \\ &= \mathbb{E}_{\mathbf{X}_v} \left[ D_{\text{KL}} \left( \mathcal{N}(\lambda_n x_n^v, \max((1 - \lambda_n)^2, \delta)) \,\|\, \mathcal{N}(0,1) \right) \right] \\ &= \mathbb{E}_{\mathbf{X}_v} \left[ \frac{1}{2} \Big( (\lambda_n x_n^v)^2 + \max((1 - \lambda_n)^2, \delta) - \log(\max((1 - \lambda_n)^2, \delta)) - 1 \Big) \right] \geq I(\mathbf{Z}_v; \mathbf{X}_v), \end{aligned} \tag{6}$$

where $\delta$ is a small positive constant (typically set to $10^{-3}$) used to prevent numerical instability by ensuring a minimum variance during training.

Following the variational treatment of mutual information proposed in (Barber & Agakov, 2004), we obtain the lower bound of $I(\mathbf{Z}_v; \mathbf{X}_t)$:

$$I(\mathbf{Z}_v; \mathbf{X}_t) = \mathbb{E}_{\mathbf{Z}_v, \mathbf{X}_t} \left[ \log \frac{P(z_v | x_t)}{Q(z_v)} \right] + \mathbb{E}_{\mathbf{X}_t} \left[ D_{\text{KL}}(Q(z_v | x_t) \,\|\, P(z_v | x_t)) \right] \geq \mathbb{E}_{\mathbf{Z}_v, \mathbf{X}_t} \left[ \log P(z_v | x_t) \right]. \tag{7}$$

To estimate the lower bound of $I(\mathbf{Z}_v; \mathbf{X}_t)$, we normalize both vectors such that $\|\mathbf{x}_t\|_2 = 1$ and $\|\mathbf{z}_v\|_2 = 1$. According to prior work (Fisher, 1953), the conditional probability density function $\log P(z_v|x_t)$ is proportional to the inner product $z_v^\top x_t$ under such normalization. Therefore, the lower bound of $I(\mathbf{Z}_v; \mathbf{X}_t)$ can be computed as:

$$\mathbb{E}_{\mathbf{Z}_v, \mathbf{X}_t} \left[\log P(z_v|x_t)\right] = \gamma \cdot \frac{1}{NM} \sum_{i=1}^{N} \sum_{j=1}^{M} (z_i^v)^\top x_j^t \leq I(\mathbf{Z}_v; \mathbf{X}_t), \tag{8}$$

where $\gamma$ is a constant scaling factor. For simplicity of computation, we set $\gamma = 1$ in our experiments.

Combining the above lower bound on $I(\mathbf{Z}_v; \mathbf{X}_t)$ with an analytically computable upper bound on $I(\mathbf{Z}_v; \mathbf{X}_v)$, we obtain the overall optimization objective of the Variational Multi-Modal Information Bottleneck:

$$\mathcal{L}_{\text{MIB}} = \beta \cdot \mathbb{E}_{\mathbf{X}_v} \left[\frac{1}{2}\left((\lambda_n x_n^v)^2 + \max((1-\lambda_n)^2, \delta) - \log(\max((1-\lambda_n)^2, \delta)) - 1\right)\right]$$
$$- \frac{1}{NM} \sum_{i=1}^{N} \sum_{j=1}^{M} (z_i^v)^\top x_j^t. \tag{9}$$

### 4.3 BINARY-GUIDED TOKEN IMPORTANCE MODELING

To enhance the discriminative capacity of the Information Bottleneck layer in distinguishing between informative and uninformative image tokens, and to facilitate subsequent clustering by promoting improved feature separability, we introduce a *Binary-Guided loss*. This loss guides the IB parameter $\lambda_n \in [0, 1]$ (defined in Eq. 3) toward a *bimodal distribution* (i.e., values near 0 or 1), thereby promoting a more discrete token filtering process. The Binary-Guided loss is defined as:

$$\mathcal{L}_{\text{BG}} = - \sum_{n=1}^{N} \left[\lambda_n \log(\lambda_n + \tau) + (1 - \lambda_n) \log(1 - \lambda_n + \tau)\right], \tag{10}$$

where $\tau = 1 \times 10^{-12}$ is a small constant for numerical stability.

Minimizing this objective encourages each token's $\lambda_n$ to polarize toward either 0 or 1, leading to a clearer "compress-or-retain" decision boundary. This not only improves the expressiveness of the retained information but also enables more effective compression of irrelevant tokens.

The total loss in our framework can be formulated as:

$$\mathcal{L}_{total} = \mathcal{L}_{CE} + \mathcal{L}_{MIB} + \mathcal{L}_{BG}. \tag{11}$$

### 4.4 ADAPTIVE IMAGE TOKEN FILTERING VIA SPECTRAL CLUSTERING

To suppress token-level redundancy, we introduce Adaptive Image Token Filtering. We argue that a fixed pruning ratio across all images is suboptimal, because content complexity varies significantly and a uniform threshold for all images risks either discarding useful information or retaining redundant features. Moreover, image tokens lie in a high-dimensional space $\mathbb{R}^d$ (where $d = 768$ in most cases) with an unknown underlying distribution. Consequently, we employ spectral clustering for a distribution-agnostic grouping.

Let $\mathbf{X}_v = [x_1^v, \dots, x_N^v] \in \mathbb{R}^{d \times N}$ denote the set of $N$ image tokens. Each $x_i^v$ is treated as a vertex in an undirected graph. The edge between $x_i^v$ and $x_j^v$ carries a weight $w_{ij} = \frac{(x_i^v)^\top x_j^v}{\|x_i^v\| \|x_j^v\|}$, yielding the adjacency matrix $\mathbf{W} = [w_{ij}] \in \mathbb{R}^{N \times N}$. The degree matrix $\mathbf{D} = \text{diag}(d_1, \dots, d_N)$ contains entries $d_i = \sum_j w_{ij}$. The normalized Laplacian is defined as:

$$\mathbf{L} = \mathbf{D}^{-1/2}(\mathbf{D} - \mathbf{W})\mathbf{D}^{-1/2}. \tag{12}$$

Compute the first $k$ eigenvectors $\mu_1, \ldots, \mu_k$ of $\mathbf{L}$ corresponding to the smallest non-zero eigenvalues, and stack them column-wise to form the matrix $\mathbf{U}$. Applying $k$-means clustering to the rows of $\mathbf{U}$ produces $k$ clusters $\mathcal{C} = \{c_1, \ldots, c_k\}$ (we use $k = 5$ as the default value).

For each cluster $c_j \in \mathcal{C}$, we compute its MIB compression term:

$$\bar{I}_j = \frac{1}{|c_j|} \sum_{z_i^v \in c_j} I[z_i^v; x_i^v]. \tag{13}$$

We identify the most redundant cluster as that with the lowest $\bar{I}_j$ and retain only tokens outside it:

$$\mathcal{I}^{\text{keep}} = \big\{ i \mid z_i^v \notin c_\varsigma, \ \ \varsigma = \arg\min_{1 \le j \le k} \bar{I}_{(j)} \big\}. \tag{14}$$

## 5 EXPERIMENTS

### 5.1 EXPERIMENTAL SETTINGS

**Image Reasoning.** To evaluate the effectiveness of VIBRA in enhancing image reasoning, we integrated VIBRA into two representative multimodal reasoning models: MM-CoT (Zhang et al., 2023b) and MC-CoT (Tan et al., 2024), conducting experiments on the ScienceQA dataset. We compared our models with a range of baselines, including UnifiedQA$_{\text{Base}}$ (Khashabi et al., 2020), UnifiedQA$_{\text{Base}}$ w/ CoT (Lu et al., 2022), LLaMA-Adapter (Zhang et al., 2023a), LaVIN-13B (Luo et al., 2023), GPT-3.5 w/ CoT (OpenAI, 2022), and GPT-4 w/ CoT (Achiam et al., 2023).

**Visual Understanding.** To assess improvements in visual understanding, we integrated VIBRA into MiniGPT-4 (Zhu et al., 2024) and LLaVA-1.5 (Liu et al., 2024b) at the 7B scale, conducting experiments on popular benchmarks,including POPE (Li et al., 2023b), MME (Chaoyou et al., 2023) and Offline POPE(OPOPE) (Chen et al., 2024b). Please refer to Appendix A.1 for the introduction of each benchmark. Our method was compared against strong baselines with greedy decoding strategies, including DOLA (Chuang et al., 2023), VCD (Leng et al., 2024), OPERA (Huang et al., 2024), HALC (Chen et al., 2024b), SID (Huo et al., 2025) and Nullu (Yang et al., 2025).

**Implementation Details.** In the image reasoning experiments, we followed the experimental setups of MM-CoT and MC-CoT. For the visual understanding experiments, all parameters of the MiniGPT-4 and LLaVA-1.5 models were frozen, and 5,000 image-text pairs were randomly sampled from LLaVA-150k (Liu et al., 2024b) to train the VIBRA module. The number of clusters in spectral clustering was set to 5. More training details are provided in Appendix A.2. For the VIBRA module, the selection of the $\beta$ value is detailed in Appendix A.7.

### 5.2 MAIN RESULTS

**ScienceQA.** To evaluate the effectiveness of VIBRA in enhancing the image reasoning capabilities of baseline models, we integrated VIBRA into a series of representative models, including MM-CoT and MC-CoT, on the ScienceQA dataset. The main results are summarized in Table 1. Specifically, MM-CoT$_{\text{T5-Base}}$ + VIBRA achieves an accuracy of 91.65%, representing a **2.92%** improvement over MM-CoT$_{\text{T5-Base}}$, while MC-CoT$_{\text{T5-Base}}$ + VIBRA achieves 92.74%, yielding a **3.87%** improvement. These results highlight VIBRA's effectiveness in strengthening multimodal reasoning performance. Notably, VIBRA brings substantial improvements on the **IMG Metric**, which specifically measures the model's performance on image-based questions. Based on the T5-Base backbone, VIBRA improves the image reasoning accuracy of MM-CoT and MC-CoT by **4.61%** and **7.24%**, respectively, demonstrating VIBRA 's ability to significantly enhance image reasoning capabilities. In addition, we evaluate the quality of the generated explanations using the RougeL metric in Appendix A.3.

**OPOPE Evaluation.** Following the experimental protocols of HALC (Chen et al., 2024b) and Nullu (Yang et al., 2025), we evaluate visual hallucination using the Offline POPE (OPOPE) benchmark. As shown in Table 4, VIBRA consistently outperforms the state-of-the-art baselines, indicating a clear reduction in hallucination. Moreover, VIBRA achieves effective hallucination mitigation when integrated into two distinct LVLM architectures, demonstrating its plug-and-play applicability across different model designs.

Table 1: The accuracy score comparison against baselines on ScienceQA dataset. Here, Size = size of the backbone model, NAT = Natural Science, SOC = Social Science, LAN = Language Science, TXT = Text context, IMG = Image context, NO = No context, G1-6 = Grade 1 to 6, G7-12 = from Grade 7 to 12.

| Model | Size | NAT | SOC | LAN | TXT | IMG | NO | G1-6 | G7-12 | Avg. |
|---|---|---|---|---|---|---|---|---|---|---|
| Human Average | - | 90.23 | 84.97 | 87.48 | 89.60 | 87.50 | 88.10 | 91.59 | 82.42 | 88.40 |
| UnifiedQA$_{Base}$ | 223M | 68.16 | 69.18 | 74.91 | 63.78 | 61.38 | 77.84 | 72.98 | 65.00 | 70.12 |
| UnifiedQA$_{Base}$ w/ CoT | 223M | 71.00 | 76.04 | 78.91 | 66.42 | 66.53 | 81.81 | 77.06 | 68.82 | 74.11 |
| LLaMA-Adapter | 6B (1.2M) | 84.37 | 88.30 | 84.36 | 83.72 | 80.32 | 86.90 | 85.83 | 84.05 | 85.19 |
| LaVIN-13B | 13B (5.4M) | 89.88 | 94.49 | 89.92 | 88.95 | 87.61 | 91.85 | 91.45 | 89.72 | 90.83 |
| GPT-3.5 w/ CoT | >175B | 75.44 | 70.97 | 78.09 | 74.68 | 67.43 | 79.93 | 78.23 | 69.68 | 75.15 |
| GPT-4 w/ CoT | >175B | 85.48 | 72.44 | 90.27 | 82.65 | 71.49 | 92.89 | 86.66 | 79.04 | 83.99 |
| MM-CoT$_{T5-Base}$ | 223M | 91.87 | 80.09 | 89.27 | 91.64 | 86.96 | 90.45 | 89.46 | 87.41 | 88.73 |
| MM-CoT$_{T5-Base}$ + **VIBRA** | 223M+2.07M | **92.76** | **90.33** | **90.45** | **92.52** | **91.57** | **91.71** | **91.89** | **91.23** | **91.65** |
| MC-CoT$_{T5-Base}$ | 223M | 91.74 | 79.30 | 90.73 | 92.23 | 86.07 | 91.36 | 89.57 | 87.61 | 88.87 |
| MC-CoT$_{T5-Base}$ + **VIBRA** | 223M+2.07M | **93.07** | **93.25** | **91.64** | **92.67** | **93.31** | **92.75** | **92.88** | **92.49** | **92.74** |

**POPE Evaluation.** To further assess VIBRA's ability to improve visual understanding, we integrated it into MiniGPT-4$_{7B}$ and evaluated it on the POPE dataset. As shown in Table 5, we observe that VIBRA yields consistent improvements across most evaluation settings. These results demonstrate that VIBRA significantly enhances the model's visual understanding capability and effectively mitigates hallucinations in vision-language tasks.

**MME Benchmark.** We evaluate on both object-level subsets ("existence" and "count") and attribute-level subsets ("position" and "color") from the MME benchmark. We report results for LLaVA-1.5$_{7B}$ as a representative LVLM. Table 6 shows that incorporating VIBRA yields consistent improvements on perception- and recognition-oriented tasks, reflecting enhanced perceptual fidelity and recognition accuracy.

## 5.3 ABLATION STUDY

To better understand the contribution of each component in VIBRA, we conduct comprehensive ablation studies on both the POPE and ScienceQA datasets. As summarized in Tables 2 and 3, we evaluate the impact of the following modules: Multi-Modal Information Bottleneck (MIB), Adaptive Image Token Filtering via Spectral Clustering (TFSC), and Binary-Guided Token Importance Modeling (BG).

Table 2: Ablation results on the POPE dataset based on the MiniGPT-4$_{7B}$. We use the fine-tuned MiniGPT-4$_{7B}$ model as our baseline.

| Model | Random $F_1 \uparrow$ | Popular $F_1 \uparrow$ | Adversarial $F_1 \uparrow$ |
|---|---|---|---|
| Baseline | 80.8 | 72.9 | 71.2 |
| + MIB | 82.5 | 74.1 | 71.8 |
| + MIB + TFSC | 83.5 | 76.2 | 73.0 |
| + MIB + TFSC + BG | **84.1** | **76.5** | **73.5** |

Table 3: Ablation results on the ScienceQA dataset based on the MM-CoT$_{T5-Base}$.

| Model | IMG |
|---|---|
| Baseline | 86.96 |
| + MIB | 87.85 (↑0.89) |
| + MIB + TFSC | 90.93 (↑3.08) |
| + MIB + TFSC + BG | **91.57 (↑0.64)** |

**Study on the Multi-Modal Information Bottleneck (MIB).** As shown in Tables 2 and 3, incorporating the Multi-Modal Information Bottleneck consistently improves model performance across both datasets. Additionally, as illustrated in Appendix A.5, we visualize the compression term from MIB to interpret its behavior. The highlighted regions indicate that MIB effectively focuses on semantically relevant areas of the input images. These results demonstrate that MIB not only improves quantitative performance but also enhances the semantic alignment between vision and language modalities.

Table 4: Experimental results of OPOPE on the MiniGPT-4[7B] and LLaVA-1.5[7B] models

| OPOPE Split | Methods | MiniGPT-4 | | | | LLaVA-1.5 | | | |
|---|---|---|---|---|---|---|---|---|---|
| | | Accuracy↑ | Precision↑ | Recall↑ | $F_1$ ↑ | Accuracy↑ | Precision↑ | Recall↑ | $F_1$ ↑ |
| Random | Greedy | 72.42 | 98.49 | 45.53 | 62.27 | 81.52 | 98.41 | 64.07 | 77.61 |
| | Beam | 72.65 | 98.70 | 45.90 | 62.66 | 81.67 | 98.67 | 64.20 | 77.79 |
| | VCD | 72.35 | 98.19 | 45.53 | 62.21 | 80.57 | 98.41 | 62.13 | 76.17 |
| | OPERA | 72.57 | 98.77 | 45.70 | 62.49 | 81.62 | 98.57 | 64.17 | 77.73 |
| | DOLA | 72.45 | 98.58 | 45.57 | 62.33 | 81.38 | 98.11 | 64.00 | 77.47 |
| | HALC | 72.08 | 98.62 | 44.80 | 61.61 | 79.58 | 98.21 | 60.27 | 74.70 |
| | Nullu | **72.68** | 99.06 | 45.80 | 62.64 | 81.18 | 98.05 | 63.63 | 77.18 |
| | **VIBRA** | 72.65 | **99.30** | **47.27** | **64.05** | **82.54** | 98.82 | **66.93** | **79.81** |
| Popular | Greedy | 70.80 | 92.01 | 45.53 | 60.92 | 78.93 | 91.17 | 64.07 | 75.25 |
| | Beam | 71.32 | 93.35 | 45.90 | 61.54 | 79.30 | 91.98 | 64.20 | 75.62 |
| | VCD | 70.33 | 90.30 | 45.53 | 60.54 | 77.57 | 89.87 | 62.13 | 73.47 |
| | OPERA | 71.10 | 92.82 | 45.70 | 61.25 | 79.22 | 91.80 | 64.17 | 75.54 |
| | DOLA | 70.90 | 92.33 | 45.57 | 61.02 | 78.72 | 90.69 | 64.00 | 75.04 |
| | HALC | 70.92 | 93.80 | 44.80 | 60.64 | 77.47 | 91.87 | 60.27 | 72.79 |
| | Nullu | 71.97 | 96.08 | 45.80 | 62.03 | 79.80 | **94.06** | 63.63 | 75.91 |
| | **VIBRA** | **72.87** | **96.86** | **47.27** | **63.53** | **80.40** | 91.68 | **66.87** | **77.33** |
| Adversarial | Greedy | 70.43 | 90.65 | 45.53 | 60.62 | 76.97 | 86.36 | 64.07 | 73.56 |
| | Beam | 70.98 | 92.06 | 45.90 | 61.26 | 77.27 | 86.92 | 64.20 | 73.85 |
| | VCD | 69.82 | 88.43 | 45.53 | 60.11 | 75.88 | 85.71 | 62.13 | 72.04 |
| | OPERA | 70.78 | 91.63 | 45.70 | 60.98 | 77.03 | 86.40 | 64.17 | 73.64 |
| | DOLA | 70.50 | 90.85 | 45.57 | 60.70 | 76.85 | 86.18 | 64.00 | 73.45 |
| | HALC | 70.52 | 92.22 | 44.80 | 60.30 | 76.57 | **89.44** | 60.27 | 72.01 |
| | Nullu | 71.10 | 92.73 | 45.80 | 61.32 | 77.58 | 88.27 | 63.63 | 73.95 |
| | **VIBRA** | **72.67** | **96.07** | **47.27** | **63.36** | **78.93** | 88.75 | **66.27** | **75.88** |

Table 5: Experimental results on POPE dataset. All baseline methods are implemented based on the MiniGPT-4[7B] model.

| Methods | Random | | Popular | | Adversarial | |
|---|---|---|---|---|---|---|
| | Accuracy↑ | $F_1$ ↑ | Accuracy↑ | $F_1$ ↑ | Accuracy↑ | $F_1$ ↑ |
| Greedy | 64.33 | 73.14 | 56.63 | 69.13 | 55.17 | 68.42 |
| Beam Search | 62.10 | 71.84 | 56.47 | 68.95 | 55.50 | 68.48 |
| DoLA | 64.27 | 72.68 | 56.58 | 68.65 | 55.85 | 68.29 |
| VCD | 57.90 | 64.73 | 55.30 | 63.92 | 52.90 | 61.61 |
| HALC | 64.87 | 73.44 | 57.00 | 69.31 | 55.53 | 68.60 |
| Nullu | 77.23 | 77.53 | 70.13 | 72.45 | **66.70** | 70.22 |
| **VIBRA** | **81.78** | **84.09** | **71.43** | **76.51** | 66.50 | **73.53** |

Table 6: Object-level and Attribute-level Hallucination Evaluations on MME dataset. All baseline methods are implemented based on the LLaVA-1.5[7B] model.

| Methods | Object-level | | Attribute-level | | Total |
|---|---|---|---|---|---|
| | Existence ↑ | Count ↑ | Position ↑ | Color ↑ | |
| Greedy | 182.3 | 130.3 | 126.8 | 155.7 | 594.1 |
| Dola | 180.1 | 127.4 | 119.3 | 154.4 | 581.2 |
| VCD | 179.5 | 128.1 | 123.8 | 155.5 | 586.9 |
| HALC | 167.7 | 121.3 | 106.7 | 150.7 | 546.4 |
| Nullu | **190.0** | 121.1 | 105.6 | 156.7 | 573.4 |
| SID | 183.9 | 132.2 | 127.8 | 155.9 | 599.8 |
| **VIBRA** | 185.0 | **145.0** | **133.3** | **165.0** | **628.3** |

**Study on Adaptive Image Token Filtering via Spectral Clustering (TFSC) and Comparison of Token Filtering Strategies.** From the results in Tables 2 and 3, we observe a significant performance improvement after introducing the TFSC module. To investigate the impact of different metrics used for token filtering, we compare our MIB Compression Term with a baseline using cosine similarity in selecting image tokens, with the number of retained tokens varying from 5 to 50. Note that 50 corresponds to using all image tokens in MM-CoT[T5-Base]. As illustrated in Figure 3, the MIB-based method significantly outperforms cosine similarity in select-

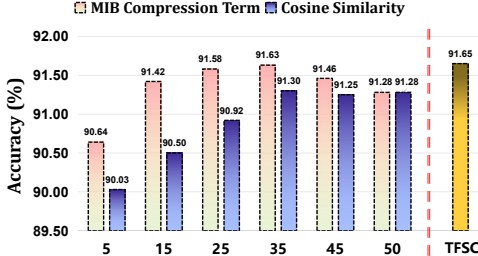

Figure 3: Comparison of token filtering strategies in MM-CoT[T5-Base] on the ScienceQA dataset.

ing useful image tokens, confirming that the MIB module provides a more robust and semantically aligned signal for guiding token selection. The figure further suggests that using a uniform pruning threshold across all images is suboptimal, as it fails to account for the varying complexity of visual content. In contrast, TFSC dynamically adapts the filtering process, preserving informative tokens while discarding redundant ones.

**Study on Binary-Guided Token Importance Modeling (BG).** As shown in Tables 2 and 3, the addition of the BG module yields consistent gains. It improves the MIB's ability to distinguish informative tokens, thereby enhancing TFSC with more accurate token importance guidance. In Appendix A.4, we provide visualizations to verify the effectiveness of BG module.

**Speed-Accuracy Analysis of VIBRA.** We conducted a comparative study on inference speed and accuracy using LLaVA-1.5₇ᴮ as the base model on the POPE dataset, with all experiments executed on a NVIDIA L40 GPU to ensure a consistent and controlled evaluation environment.

Table 7: Comparison of inference speed and POPE $F_1$ scores across different methods.

| Methods | Time (s/token) | POPE-Random | POPE-Popular | POPE-Adversarial | POPE-Average |
|---|---|---|---|---|---|
| VCD | 0.127 | 89.350 | 86.409 | 81.022 | 85.593 |
| OPERA | 0.136 | 88.933 | 86.297 | 80.923 | 85.384 |
| SID | 0.137 | 89.042 | 85.961 | 81.427 | 85.476 |
| HALC | 0.283 | 81.650 | 75.130 | 73.401 | 76.727 |
| VIBRA-MIB | 0.061 | 87.229 | 86.149 | 84.035 | 85.804 |
| VIBRA-MIB+TFSC($k = 3$) | 0.073 | 87.787 | 86.135 | 84.026 | 85.983 |
| VIBRA-MIB+TFSC($k = 5$) | 0.074 | 88.274 | 86.904 | **84.232** | 86.470 |
| VIBRA-MIB+TFSC($k = 7$) | 0.075 | 87.968 | 86.407 | 84.033 | 86.136 |
| VIBRA-MIB+TFSC($k = 9$) | 0.081 | **88.736** | **87.104** | 84.167 | **86.669** |

As shown in Table 7, VIBRA-MIB+TFSC ($k = 9$) demonstrates both efficiency and effectiveness, reducing the per-token inference time to **0.081 seconds**—approximately **3.5×** faster than HALC (**0.283 seconds**)—while simultaneously improving the POPE average $F_1$ score by **9.94%**. These findings demonstrate that, compared to typical decoder-based methods, VIBRA can more effectively suppress hallucinations while substantially reducing resource consumption. Furthermore, we analyzed the additional overhead introduced by the MIB and TFSC modules. The experiments indicate that incorporating TFSC on top of MIB (e.g., with $k = 9$) increases the inference time by only **0.02 seconds**, while improving the average $F_1$ score by **0.86%**. This confirms that VIBRA achieves further performance gains with a marginal and acceptable computational cost.

**Comparison with Token-Pruning Baselines.** We conducted comparative experiments using the LLaVA-v1.5-7B backbone and evaluated performance on the POPE dataset with POPE Average $F_1$ score as the metric. Simignore, PruMerge, and FastV were evaluated under the same backbone and identical image-token retention ratios (retain $k\%$). As shown in Table 8, VIBRA's advantage becomes particularly evident as the retention ratio declines (60%–80%), indicating that VIBRA performs better under high compression.

Table 8: Performance comparison under different proportions of retained image tokens.

| Retain $k\%$ image tokens | FastV | PruMerge | Simignore | VIBRA |
|---|---|---|---|---|
| 90% | **85.836** | 71.767 | 85.657 | 85.303 |
| 80% | 85.439 | 71.486 | 85.667 | **85.876** |
| 70% | 85.017 | 70.748 | 85.533 | **85.951** |
| 60% | 84.308 | 69.897 | 85.489 | **86.145** |

## 6 CONCLUSION

We proposed VIBRA, a plug-and-play framework that explicitly disentangles redundancy suppression across feature and token levels in vision-language processing. At the *feature-level*, our Variational Multi-Modal Information Bottleneck learns a semantic-aware gate that selectively amplifies image features aligned with the textual query while injecting controlled noise into irrelevant regions. Complementarily, at the *token-level*, the proposed Adaptive Image Token Filtering via Spectral Clustering groups visual tokens into semantically coherent subgraphs and removes those exhibiting high compression ratios—i.e., tokens that contribute minimally to the overall information budget. Furthermore, our Binary-Guided loss explicitly encourages latent gate values to converge toward near-binary decisions, making it trivial to distinguish informative from non-informative tokens and further accelerating filtering decisions. Extensive experiments demonstrated consistent accuracy gains and reduced hallucination on both reasoning-oriented and faithful-understanding tasks.

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

# A APPENDIX

## A.1 BENCHMARKS

**ScienceQA.** ScienceQA (Lu et al., 2022) is a multimodal dataset designed for scientific question answering, featuring annotated answers with detailed reasoning and explanations. It contains 21,208 multiple-choice questions across a broad range of topics, covering 3 subjects, 26 topics, 127 categories, and 379 distinct skills.

**POPE.** POPE (Li et al., 2023b) formulates the evaluation of visual hallucination as a binary classification task (yes/no) focused on object existence. The dataset consists of 500 images randomly sampled from MSCOCO (Lin et al., 2014). The full POPE evaluation includes three subsets: Random, Popular, and Adversarial, where the missing object is selected randomly, from the most frequent object list, or based on high semantic correlation with existing objects in the image, respectively. We use the $F_1$ score—the harmonic mean of precision and recall—as the evaluation metric.

**OPOPE.** OPOPE (Chen et al., 2024b) extends POPE by pre-computing candidate objects and corresponding question-answer pairs , enabling offline evaluation without runtime question generation. In this setting, the model's own outputs serve as the source for constructing yes/no questions, which are then evaluated following the POPE metrics.

**MME.** MME (Chaoyou et al., 2023) is a comprehensive benchmark for multimodal large language models that measures both perception and cognition across 14 subtasks. To reduce data-leakage and prompt-engineering bias, MME uses manually designed instruction–answer pairs and concise templates to enable fair, quantitative cross-model comparisons.

## A.2 DETAILED EXPERIMENTAL SETTINGS

In the image reasoning experiments, we followed the experimental setup of MM-CoT and MC-CoT. We adopted a two-stage framework consisting of a rationale generation phase and an answer prediction phase. Both stages were built upon a shared T5 encoder-decoder architecture (Raffel et al., 2020), with visual features extracted using the CLIP-ViT model to incorporate image information into the reasoning process. We employed UnifiedQA models (Khashabi et al., 2020) with 223M parameters as our default base backbones. The learning rate was set to $5 \times 10^{-5}$, and the maximum sequence length during training was 512 tokens.

In the visual understanding experiments, we used Vicuna-7B as the language model backbone in MiniGPT-4. We randomly sampled 5,000 image-text pairs from LLaVA-150k (Liu et al., 2024b) to train the VIBRA module. The learning rate was set to $3 \times 10^{-5}$, and the maximum training sequence length was also set to 512 tokens.

All experiments were conducted on NVIDIA GeForce RTX 4090 GPU.

## A.3 ROUGEL EVALUATION

We evaluate the quality of the generated explanations using the RougeL metric, following the evaluation protocol used in (Zhang et al., 2023b). Table 9 reports the results of answer and rationale generation on the ScienceQA dataset. VIBRA leads to an improvement of 0.73% in RougeL for MM-CoT$_{\text{T5-Base}}$ and 0.97% for MC-CoT$_{\text{T5-Base}}$, along with a consistent increase in average accuracy. These results suggest that VIBRA enhances the quality of the generated rationales.

Table 9: The RougeL score of the generated rationales and the average accuracy of predicted answers on the ScienceQA dataset.

| Model | RougeL | Avg. Acc. |
|---|---|---|
| MM-CoT$_{\text{T5-Base}}$ | 97.18 | 88.73 |
| MM-CoT$_{\text{T5-Base}}$ + **VIBRA** | **97.91** | **91.65** |
| MC-CoT$_{\text{T5-Base}}$ | 97.21 | 88.87 |
| MC-CoT$_{\text{T5-Base}}$ + **VIBRA** | **98.18** | **92.74** |

## A.4 VISUALIZATION OF THE DISTRIBUTION OF $\lambda_n$: THE POLARIZING EFFECT OF BINARY-GUIDED LOSS

To verify the effectiveness of our proposed Binary-Guided loss $\mathcal{L}_{\text{BG}}$, we visualize the distribution of the learned multi-modal information bottleneck parameter $\lambda_n$ across all image tokens. As illustrated in Figure 4, without the guidance of $\mathcal{L}_{\text{BG}}$, the $\lambda_n$ values tend to cluster around the middle range, lacking a clear decision boundary for token selection. In contrast, after applying $\mathcal{L}_{\text{BG}}$, the distribution of $\lambda_n$ becomes evidently bimodal, with values pushed toward either 0 or 1. This polarization aligns with our design goal of encouraging more discrete and interpretable filtering of visual tokens. This result empirically confirms that $\mathcal{L}_{\text{BG}}$ effectively enhances the discreteness of token filtering.

## A.5 SALIENCY MAP VISUALIZATION

We present additional visualizations of saliency maps derived from the MIB compression term in Figure 5.

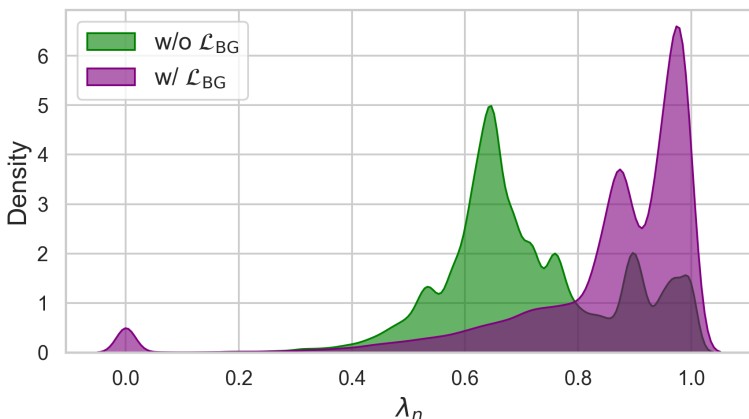

Figure 4: Distribution of the learned MIB parameter $\lambda_n$ with and without the proposed Binary-Guided loss $\mathcal{L}_{\text{BG}}$.

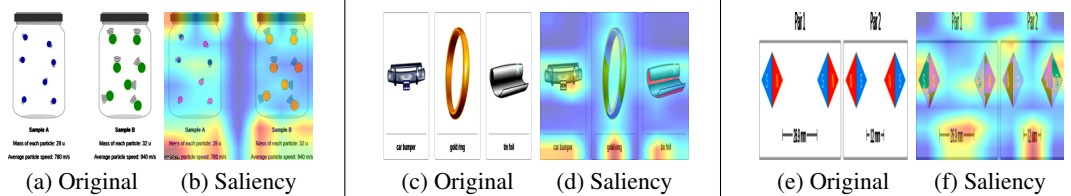

(a) Original     (b) Saliency      (c) Original     (d) Saliency      (e) Original     (f) Saliency

Figure 5: Visualization of original images and their corresponding saliency maps. Each image corresponds to a textual input: (a-b)"Question: Compare the average kinetic energies of the particles in each sample. Which sample has the higher temperature? Options: (1) neither (2) sample A (3) sample B"; (c-d)"Question: Which property do these three objects have in common? Options: (1) rough (2) shiny (3) fracile "; (e-f)"Question: Think about the magnetic force between the magets in each pair. Which of the following statements is true? Options: (1) The magnitude of the magnetic force is the same in both pairs. (2) The magnitude of the magnetic force is smaller in Pair 2. (3) The magnitude of the magnetic force is smaller in Pair 1".

## A.6 VALIDATING THE NECESSITY OF LAST-CLUSTER REMOVAL

We further demonstrate through quantitative experiments that the removed clusters indeed contain minimal valuable information. To verify whether the filtered clusters contain useful content, we employed LLaVA-v1.5 to compute the **average MIB Compression Score** $\bar{I}_j = \frac{1}{|c_j|} \sum_{z_i^v \in c_j} I\left[z_i^v; x_i^v\right]$ for each cluster on **ScienceQA** and **POPE**. This metric provides a quantitative measure of the information content of tokens within a cluster.

Table 10: Cluster values across datasets.

| Datasets | Cluster 1 | Cluster 2 | Cluster 3 | Cluster 4 | Cluster 5 |
|----------|-----------|-----------|-----------|-----------|-----------|
| POPE | 2.2367 | 1.7150 | 1.3712 | 1.0888 | **0.0346** |
| ScienceQA | 6.0228 | 1.5263 | 1.1738 | 1.1105 | **0.0297** |

As shown in Table 10, we observe that the penultimate cluster exhibits an average information content **31.47–37.39 times higher** than the last cluster. This is reasonable, as spectral clustering tends to separate informative target content from redundant background information. The redundant background tokens are grouped into a single cluster, which inherently has the lowest information content. Therefore, by filtering out the cluster with minimal information, we can effectively remove redundant image tokens.

Although **ScienceQA** and **POPE** correspond to fundamentally different tasks, both datasets consistently show that the last cluster has extremely low information content. This indicates that the phenomenon is general rather than incidental. Furthermore, our ablation studies further validate the effectiveness of the **Adaptive Image Token Filtering** module: removing the low-information cluster leads to a significant performance improvement, demonstrating that this design is both reasonable and necessary.

### A.7 HYPERPARAMETER SENSITIVITY EXPERIMENTAL ANALYSIS

**Sensitivity of the $\beta$.** To assess the sensitivity of the MIB coefficient $\beta$, we conducted a logarithmic sweep from $10^{-5}$ to 1 on the POPE dataset and evaluated the model using the POPE-Average metric. Experiments were performed on both MiniGPT-4-7B and LLaVA-v1.5-7B to examine the stability and transferability of $\beta$ across different backbones. The results are summarized in Table 10.

Table 11: Performance sensitivity to $\beta$ on different models.

| $\beta$ | $10^{-5}$ | $10^{-4}$ | $10^{-3}$ | $10^{-2}$ | $10^{-1}$ | **1** |
|---------|-----------|-----------|-----------|-----------|-----------|-------|
| Minigpt-4-7B | 78.034 | 78.065 | 78.331 | 77.801 | 75.599 | 67.345 |
| LLaVA-v1.5-7B | 85.556 | 86.022 | 85.998 | 86.148 | 85.165 | 72.572 |

Across both backbones, we observe that the performance remains highly stable within a broad range of $\beta \in [10^{-5}, 10^{-2}]$, with fluctuations below 1%. This indicates that our method is insensitive to the choice of $\beta$ within this interval. When $\beta$ becomes excessively large (e.g., $\beta \geq 10^{-1}$), the performance drops noticeably on both models, due to over-compression of image tokens under a dominant information-bottleneck constraint.

Importantly, the two backbones exhibit consistent trends, suggesting a transferable rule-of-thumb: a stable plateau exists for $\beta$ in $[10^{-5}, 10^{-2}]$, while overly large values lead to degradation.

As shown in Figure 6, we evaluate MM-CoT$_{\text{T5-Base}}$ + VIBRA on the ScienceQA dataset with $\beta$ ranging from 0.03 to 0.07. The model achieves the highest accuracy at $\beta = 0.05$, indicating that a moderate degree of compression provides the best trade-off between retaining informative features and suppressing redundancy.

**Sensitivity of the Binary-Guided loss weight.** To evaluate the sensitivity of the Binary-Guided (BG) loss, we conducted a sweep over a wide range of loss weights, from disabling the BG loss (weight = 0) to imposing a strong binary constraint (weight = 10). The experiment was conducted using the LLaVA-v1.5-7B backbone on the POPE dataset. The results are summarized in the Table 12.

We observe the following trends. First, the model remains highly stable within the range BG weight $\in [0.01, 1]$, with performance fluctuations below 0.5%. Moderate BG weighting consistently improves performance, suggesting that the BG loss provides useful guidance in suppressing visually irrelevant tokens.

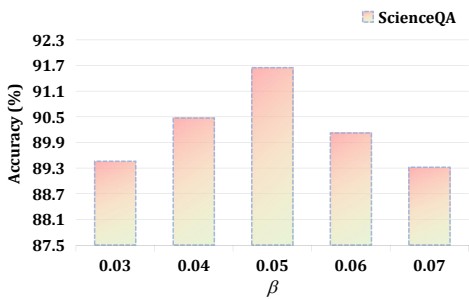

Figure 6: Effect of the compression term weight $\beta$ on model accuracy. Experiments are conducted on ScienceQA using the MM-CoT$_{\text{T5-Base}}$ + VIBRA.

Table 12: POPE-Average performance under different Binary-Guided loss weights.

| Binary-Guided loss weight | 0 | 0.01 | 0.1 | 0.5 | 1 | 5 | 10 |
|---|---|---|---|---|---|---|---|
| POPE-Average | 86.231 | 86.281 | 86.323 | 86.462 | 86.669 | 76.604 | 32.691 |

Second, when the BG weight becomes excessively large (e.g., $\geq 5$), performance drops sharply—from 86.669 at weight 1 to 76.604 (weight 5), and further to 32.691 (weight 10). This degradation is expected, as an overly strong binary constraint disrupts the smoothness and learnability of token importance distributions, ultimately harming visual-text alignment.

Overall, the results indicate that the BG loss possesses a clear and robust stability region within $[0.01, 1]$, where performance remains consistently strong and insensitive to the exact weight value.

**Sensitivity of the $\delta$.** The experiment was conducted using the LLaVA-v1.5-7B backbone to evaluate the sensitivity of the stabilization constant $\delta$.

Table 13: POPE-Average performance under different $\delta$ values.

| $\delta$ | $10^{-6}$ | $10^{-5}$ | $10^{-4}$ | $10^{-3}$ | $10^{-2}$ | $10^{-1}$ |
|---|---|---|---|---|---|---|
| POPE-Average | 86.225 | 86.324 | 86.409 | 86.669 | 86.223 | 85.855 |

As shown in Table 13, we observe that setting $\delta = 0.1$ noticeably degrades model accuracy, indicating that an excessively large stabilization term can interfere with effective learning. In contrast, when $\delta \leq 10^{-2}$, the model performance consistently remains above 86.2, forming a stable plateau across several orders of magnitude. This demonstrates that the method is not sensitive to $\delta$ within a reasonable range, and $\delta = 10^{-3}$ provides a good balance between stability and performance.

# B  LLM USAGE

We employed large language models (LLMs) to polish this manuscript. Specifically, an LLM enhanced linguistic precision, readability, and clarity through sentence rephrasing, grammar correction, and improved textual flow.

The LLM was not involved in the research methodology, experimental design, or data analysis; these elements were independently developed by the authors.

The authors assume full responsibility for the manuscript's content, ensuring that LLM-assisted text adheres to ethical guidelines and avoids plagiarism or academic misconduct.

