# OpenReview forum: "VIBRA: Redundancy-Aware Information Bottleneck for Hallucination-Resistant Vision-Language Models"
_ICLR.cc/2026/Conference — Submitted to ICLR 2026_

### Official Review · Reviewer_qNrU · 2025-10-23

**Soundness:** 1
**Presentation:** 1
**Contribution:** 2
**Rating:** 2
**Confidence:** 5

**Summary:**

The paper is motivated by the assumption that images processed by VLMs contain a lot of **unnecessary and redundant semantics, and the authors claim that it is the core reason leading to hallucinations**. To address this, the authors propose a Multi-model Information Bottleneck (MIB) module that filters out redundant information in image tokens and retains only text-relevant features. Tokens that pass through MIB are then clustered, and the largest cluster—assumed to be redundant—is removed. The core idea is that this reduces visual hallucination.

**Strengths:**

Despite the drawbacks discussed below, the Multi-modal Information Bottleneck is a solid contribution. Compared to simple image-token filtering, it offers a good technique for distilling text-relevant information, and the results are promising.

**Weaknesses:**

- **(Very major)** The paper’s motivation is that unnecessary and redundant information in a given image causes hallucinations. However, the only supporting evidence is a single simple experiment in Figure 1, and removing tokens changes performance by merely 0.3–0.4%. The core motivation of this paper lacks sufficient grounding, which is a critical issue.

- **(Very major)** On ScienceQA, VIBRA is combined on top of MM-CoT and MC-CoT. Since these CoT-style methods already outperform the baseline, claiming further gains by adding VIBRA makes a fair comparison impossible. All baselines should either be equipped with MM-CoT, or VIBRA should be paired with a simple CoT to unify the experimental setting.

- **(Major)** While criticizing prior methods for using a fixed token filter ratio (and thus lacking adaptability), VIBRA fixes the number of clusters to $K = 5$ for spectral clustering. It then treats the largest cluster as “visual redundancy” and removes it. This raises questions: that cluster could contain important information, and the second-largest cluster might be of comparable size. Overall, the process is coarse, yet there is no analysis or justification.

- **(Major ~ Minor)** Although the paper critiques the inference cost of prior work, it provides no analysis of VIBRA’s additional cost. VIBRA also incurs overhead for computing the information bottleneck, clustering, pruning, and then recomputing cross-attention with text tokens.

**Questions:**

- Since the MIB Compression Term is proposed around Intro line 62, it would be helpful to include at least a brief introduction to it there.

- In Table 1, the model-size reporting is unfair. Because these are plug-and-play methods, both the backbone model parameter count and the adapter-module parameter count should be reported for fair comparison.

---

> ### Author Response · Authors · 2025-11-25
> **Response to Reviewer qNrU (Part 1/3)**
>
> We sincerely thank you for your valuable and insightful comments, which have greatly helped us improve our manuscript. Below, we provide detailed responses to each issue you raised.
>
> > **W1.The paper’s motivation is that unnecessary and redundant information ... is a critical issue.**
>
> To address the concern that the empirical evidence for our motivation, we conducted further, more systematic experiments to strengthen the reliability and generality of our motivation.
>
> The experiment in Figure 1 is based on the CLIP-T5 model. To verify whether the same phenomenon holds across different architectures, we further evaluate another mainstream multimodal model, LLaVA-v1.5-7B, and adopt the POPE hallucination detection benchmark, which directly reflects hallucination-related issues.
>
> We continue to use cosine similarity to select semantically relevant image tokens and retain different proportions of tokens for inference. The results are as follows:
>
> | Retain k% image tokens based on cosine similarity | POPE-Random | POPE-Popular | POPE-Adversarial | POPE-Average |
> | :--- | :---: | :---: | :---: | :---: |
> | 100% (llava-v1.5-7B) | 86.372 | 85.546 | 83.339 | 85.086 |
> | 90% | 86.458 | 85.441 | 83.447 | 85.115 |
> | 80% | **86.814** | **85.767** | **83.864** | **85.481** |
> | 70% | 86.424 | 85.504 | 83.662 | 85.197 |
> | 60% | 86.241 | 85.609 | 83.410 | 85.087 |
>
> From the table, we can observe that even with a very simple token selection strategy based solely on cosine similarity, retaining 80\% of the image tokens yields a 0.4\% improvement on POPE-Average compared to the original model. This phenomenon consistently appears in both the Image Reasoning task of CLIP-T5 and the Visual Understanding task of LLaVA-V1.5-7B, indicating that the presence of redundant image tokens is a reproducible property across different models and tasks.
>
> However, because cosine similarity is itself a very coarse heuristic, the overall performance gains remain limited (approximately 0.3\%--0.4\%). This further reinforces our motivation: simple heuristics are insufficient for reliably identifying semantically critical tokens. Therefore, a more principled multimodal information bottleneck mechanism is needed to enable adaptive and effective token filtering.
>
> To validate this hypothesis, we conducted fixed-ratio (fixed $k$\%) token selection experiments based on our VIBRA module. Unlike cosine similarity, we used the compression term in the multimodal information bottleneck(MIB) as the measure of token importance, and compared model performance under identical token-retention ratios $k$.
>
> | Retain k% image tokens based on MIB | POPE-Random | POPE-Popular | POPE-Adversarial | POPE-Average |
> | :--- | :---: | :---: | :---: | :---: |
> | 100% (llava-v1.5-7B) | 86.372 | 85.546 | 83.339 | 85.086 |
> | 90% | 86.753 | 85.788 | 83.369 | 85.303 |
> | 80% | 87.159 | 86.606 | 83.864 | 85.876 |
> | 70% | 87.679 | 86.283 | 83.890 | 85.951 |
> | 60% | 87.495 | 86.717 | 84.225 | 86.145 |
> | **Adaptive token filtering** | **88.736** | **87.104** | **84.167** | **86.669** |
>
> From the table, we observe that under the same token-retention ratio $k\%$, the VIBRA MIB-based metric consistently outperforms cosine similarity on POPE-Average. This indicates that the compression term captures semantic information more accurately and can more effectively distinguish critical tokens from redundant ones.
>
> Under the **Adaptive token Filtering** setting, the model no longer requires a fixed $k$, but instead adaptively determines the number of tokens to retain for each image. This leads to a POPE-Average F1 score of **86.669**, representing a **+1.583\%** improvement over the original LLaVA-v1.5-7B F1 score of **85.086**.
>
> These results further reinforce our central claim: **images indeed contain redundant tokens, and such redundancy increases the model’s tendency to hallucinate**. In contrast, applying a multimodal information bottleneck for adaptive token filtering can significantly reduce hallucination behaviors.

---

> ### Author Response · Authors · 2025-11-25
> **Response to Reviewer qNrU (Part 2/3)**
>
> > **W2. On ScienceQA ... VIBRA should be paired with a simple CoT to unify the experimental setting.**
>
> We sincerely thank the reviewer for their valuable feedback. We re-conducted our experiments to eliminate confounding factors introduced by the complex CoT strategies used in MM-CoT and MC-CoT. Specifically, we retrained VIBRA under a simpler, single-stage CoT setup. Unlike MM-CoT (a two-stage multimodal CoT) or MC-CoT (a two-stage voting-based CoT) which employ elaborate multi-stage reasoning processes, our approach uses a straightforward single-stage CoT strategy: during training, the model directly outputs both the reasoning and the answer, without multi-stage inference or a voting mechanism.
>
> | Methods | NAT | SOC | LAN | TXT | IMG | NO | G1-6 | G7-12 | Avg. |
> | :--- | :---: | :---: | :---: | :---: | :---: | :---: | :---: | :---: | :---: |
> | MM-COT | **91.87** | 80.09 | 89.27 | 91.64 | 86.96 | 90.45 | 89.46 | 87.41 | 88.73 |
> | MC-COT | 91.74 | 79.30 | 90.73 | **92.23** | 86.07 | **91.36** | 89.57 | 87.61 | 88.87 |
> | VIBRA w/o. COT | 86.94 | **97.98** | **92.64** | 88.91 | **92.02** | 88.85 | **92.36** | **87.80** | **90.73** |
>
> As shown in the table, even without any advanced CoT techniques, VIBRA still outperforms MM-CoT by **+2.00\%** and MC-CoT by **+1.86\%**. This demonstrates that the performance gains do not stem from stronger CoT training, but are indeed attributable to VIBRA itself.
>
> > **W3. While criticizing prior methods for using a fixed ... yet there is no analysis or justification.**
>
> We further demonstrate through quantitative experiments that the removed clusters indeed contain minimal valuable information. To verify whether the filtered clusters contain useful content, we employed LLaVA-v1.5 to compute the **average MIB Compression Score** $\bar{I} _ {j}= \frac{1}{\lvert c _ {j}\rvert} \sum _ {z _ {i}^{v}\in c_{j}} I\bigl[z _ {i}^{v};x _ {i}^{v}\bigr]$ for each cluster on **ScienceQA** and **POPE**. This metric provides a quantitative measure of the information content of tokens within a cluster. The experimental results are presented below.
>
> | datasets | Cluster 1 | Cluster 2 | Cluster 3 | Cluster 4 | Cluster 5 |
> | :--- | :---: | :---: | :---: | :---: | :---: |
> | POPE | 2.2367 | 1.7150 | 1.3712 | 1.0888 | **0.0346** |
> | ScienceQA | 6.0228 | 1.5263 | 1.1738 | 1.1105 | **0.0297** |
>
> We observe that the penultimate cluster exhibits an average information content **31.47--37.39 times higher** than the last cluster. This is reasonable, as spectral clustering tends to separate informative target content from redundant background information. The redundant background tokens are grouped into a single cluster, which inherently has the lowest information content. Therefore, by filtering out the cluster with minimal information, we can effectively remove redundant image tokens.
>
> Although **ScienceQA** and **POPE** correspond to fundamentally different tasks, both datasets consistently show that the last cluster has extremely low information content. This indicates that the phenomenon is general rather than incidental. Furthermore, our ablation studies further validate the effectiveness of the **Adaptive Image Token Filtering** module: removing the low-information cluster leads to a significant performance improvement, demonstrating that this design is both reasonable and necessary.

---

> ### Author Response · Authors · 2025-11-25
> **Response to Reviewer qNrU (Part 3/3)**
>
> > **W4. Although the paper critiques the inference cost of prior work ... then recomputing cross-attention with text tokens.**
>
> In our paper, we highlight that many prior "decoder-based methods'' incur substantial computational overhead during inference, whereas VIBRA achieves a significant reduction in hallucination while also lowering resource consumption. To substantiate this claim, we conducted a comparative study on inference speed and accuracy using LLaVA-v1.5-7B as the base model on the POPE dataset, with all experiments executed on a NVIDIA L40 GPU to ensure a consistent and controlled evaluation environment. The results are summarized as follows.
>
> | Methods | Time(s/token) | POPE-Random | POPE-Popular | POPE-Adversarial | POPE-Average |
> | :--- | :---: | :---: | :---: | :---: | :---: |
> | VCD | 0.127 | 89.350 | 86.409 | 81.022 | 85.593 |
> | OPERA | 0.136 | 88.933 | 86.297 | 80.923 | 85.384 |
> | SID | 0.137 | 89.042 | 85.961 | 81.427 | 85.476 |
> | HALC | 0.283 | 81.650 | 75.130 | 73.401 | 76.727 |
> | VIBRA-MIB | 0.061 | 87.229 | 86.149 | 84.035 | 85.804 |
> | VIBRA-MIB+TFSC($k=3$) | 0.073 | 87.787 | 86.135 | 84.026 | 85.983 |
> | VIBRA-MIB+TFSC($k=5$) | 0.074 | 88.274 | 86.904 | 84.232 | 86.470 |
> | VIBRA-MIB+TFSC($k=7$) | 0.075 | 87.968 | 86.407 | 84.033 | 86.136 |
> | VIBRA-MIB+TFSC($k=9$) | 0.081 | **88.736** | **87.104** | **84.167** | **86.669** |
>
> **VIBRA-MIB+TFSC ($k=9$)** demonstrates both efficiency and effectiveness, reducing the per-token inference time to **0.081 seconds**—approximately **3.5$\times$** faster than HALC (**0.283 seconds**)—while simultaneously improving the POPE average accuracy by **9.94%**. These findings demonstrate that, compared to typical decoder-based methods, VIBRA can more effectively suppress hallucinations while substantially reducing resource consumption.
>
> Furthermore, we analyzed the additional overhead introduced by the MIB and TFSC modules. The experiments indicate that incorporating TFSC on top of MIB (e.g., with $k=9$) increases the inference time by only **~0.02 seconds**, while improving the average F1 score by **0.86**. This confirms that VIBRA achieves further performance gains with a marginal and acceptable computational cost.
>
> > **Q1.Since the MIB Compression Term is proposed ... introduction to it there.**
>
> Thank you for the suggestion. A brief introduction to the MIB Compression Term has already been provided in the Introduction section.
>
>
> > **Q2.In Table 1, the model-size reporting ... for fair comparison.**
>
> The additional parameters introduced by the Multi-Modal Information Bottleneck (MIB) module are 2.07M, and this has been reflected in the revised Table 1 to ensure a fair comparison.

---

### Official Review · Reviewer_51Sk · 2025-10-28

**Soundness:** 3
**Presentation:** 3
**Contribution:** 2
**Rating:** 4
**Confidence:** 3

**Summary:**

This paper tackles visual hallucination in Vision–Language Models (VLMs) by reducing redundant and noisy image features, which the authors identify as a primary cause of hallucination. They propose VIBRA, a plug‑and‑play module that filters redundant visual features via a multimodal information bottleneck (MIB) objective and adaptive token filtering using spectral clustering with compression‑aware pruning. In addition, they introduce a Binary‑Guided (BG) loss that encourages near‑binary separation between informative and noisy features, improving visual information gating. In experiments, the method enhances visual reasoning and reduces hallucinations across several VLM architectures.

**Strengths:**

S1. Each component of VIBRA is described clearly and concisely. The MIB objective is well derived, with appropriate references supporting the upper bound on $ I(Z_v; X_v) $ and the lower bound on $ I(Z_v; X_t) $.

S2. The ablation studies support the contribution of each component (MIB, TFSC, and BG; Sec. 5.3). In addition, the distribution of the learned gating parameter $ \lambda_n $ with and without the proposed BG loss supports the validity of the objective.

S3. The authors provide qualitative results by visualizing the original images and their saliency maps derived from the MIB compression term (App. A.6).

**Weaknesses:**

W1. **Marginal novelty.**
The use of the information bottleneck for hallucination mitigation has prior art. For example, variational information bottleneck has been introduced for reducing object hallucination [1]. The paper’s novelty lies more in the system‑level combination (MIB + spectral clustering with compression‑aware pruning + BG loss) than in fundamentally new primitives.

W2. **Lack of token‑pruning baselines.**
The current comparisons focus on decoding‑time mitigation (e.g., VCD, OPERA, HALC, SID). Because the proposed method performs token filtering, pruning/merging‑based baselines should be included for direct head‑to‑head comparison under the same backbone and token budget (e.g., SimIgnore, PuMer, PruMerge, FastV).

W3. **Missing efficiency report.**
While the paper highlights a benefit of “no extra decoding stages,” the efficiency side is not yet quantified. It would strengthen the work to include an accuracy–efficiency Pareto—e.g., per‑image wall‑clock latency (ms) and images/sec—with the module on/off and across different  $k$ values (same hardware/backbone).

[1]:  Bai et al., Mitigating Hallucinations in Large Vision-Language Models by Adaptively Constraining Information Flow. Proceedings of the AAAI Conference on Artificial Intelligence, 2025.

**Questions:**

Please see the weaknesses above.

---

> ### Author Response · Authors · 2025-11-27
> **Response to Reviewer 51Sk (Part 1/2)**
>
> We sincerely thank you for your valuable and insightful comments, which have greatly helped us improve our manuscript. Below, we provide detailed responses to each issue you raised.
>
> > **W1. Marginal novelty.**
>
> Thank you for the insightful comments. We agree that both AdaVIB[1] and our method VIBRA are inspired by the information-bottleneck principle; however, they differ in several essential aspects of design, objective formulation, and implementation, which lead to fundamentally different mechanisms and outcomes:
>
> 1. **Different supervision objectives.**
> AdaVIB is trained under a standard cross-entropy objective, whereas **VIBRA directly optimizes the mutual information between the compressed visual representation and the textual features**. This makes VIBRA a genuinely **multimodal information bottleneck** whose objective explicitly encourages the preservation of information that is synergistically relevant to the textual semantics, thereby suppressing hallucination at its root.
>
> 2. **Different noise injection strategies and compression granularity.**
> AdaVIB injects Gaussian noise into a global visual representation. In contrast, VIBRA performs **feature-level** compression by interpolating each image token between its original feature and Gaussian noise via a learnable $\lambda$ , and additionally performs **token-level** compression using TFSC to remove the most redundant tokens. This dual-granularity design provides substantially stronger redundancy reduction.
>
> 3. **Discrete decision making and interpretability.**
> VIBRA introduces a **Binary-Guided Loss** that encourages $\lambda$ to become nearly binary, effectively producing hard keep/drop decisions. This improves the decisiveness of hallucination mitigation and further enhances the model's interpretability, as illustrated by the $\lambda$ distribution visualizations in Appendix A.6.
>
> 4. **Empirical differences.**
>
>     | Method | POPE-Random | POPE-Popular | POPE-Adversarial | POPE-Average |
>     |--------|-------------|--------------|------------------|--------------|
>     | AdaVIB  | 83.95       | 75.13        | 72.54            | 77.21        |
>     | VIBRA   | **84.09**       | **76.51**        | **73.53**            | **78.04**        |
>
>     | Method | OPOPE-Random | OPOPE-Popular | OPOPE-Adversarial | OPOPE-Average |
>     |--------|--------------|---------------|-------------------|---------------|
>     | AdaVIB  | 62.98        | 62.63         | 62.29             | 62.63         |
>     | VIBRA   | **64.05**        | **63.53**         | **63.36**             | **63.65**         |
>
> On the MiniGPT-4-7B backbone, VIBRA achieves an average F1 of 78.04 on POPE (outperforming AdaVIB by **0.83**) and 63.65 on the more challenging OPOPE benchmark (a gain of **1.02**).
>
>
>
> In summary, VIBRA contributes a **multi-modal, dual-granularity information bottleneck framework with discrete decision mechanisms**, which goes beyond a simple adaptation or incremental extension of existing methods such as AdaVIB.
>
> [1]: Bai et al., Mitigating Hallucinations in Large Vision-Language Models by Adaptively Constraining Information Flow. Proceedings of the AAAI Conference on Artificial Intelligence, 2025.
>
> > **W2. Lack of token‑pruning baselines.**
>
> Thank you for your suggestion. We conducted comparative experiments using the LLaVA-v1.5-7B backbone and evaluated performance on the POPE dataset with POPE-Average F1 as the metric. Because PuMer's public implementation has not released a reproduction for the LLaVA-v1.5-7B backbone, we were unable to include it under the same backbone and therefore did not compare against it. The remaining methods (Simignore, PruMerge, and FastV) were evaluated under the same backbone and identical image-token retention ratios (retain k%), and the results are reported below.
>
> | Retain k% image tokens | FastV  | PruMerge | Simignore | VIBRA  |
> |-----------------------|--------|----------|-----------|--------|
> | 90%                   | **85.836** | 71.767   | 85.657    | 85.303 |
> | 80%                   | 85.439 | 71.486   | 85.667    | **85.876** |
> | 70%                   | 85.017 | 70.748   | 85.533    | **85.951** |
> | 60%                   | 84.308 | 69.897   | 85.489    | **86.145** |
>
> Our experiments demonstrate that VIBRA's advantage becomes particularly evident as the retention ratio declines (60%–80%), indicating that VIBRA performs better under high compression.

---

> ### Author Response · Authors · 2025-11-27
> **Response to Reviewer 51Sk (Part 2/2)**
>
> > **W3. Missing efficiency report.**
>
> In our paper, we highlight that many prior "decoder-based methods'' incur substantial computational overhead during inference, whereas VIBRA achieves a significant reduction in hallucination while also lowering resource consumption. To substantiate this claim, we conducted a comparative study on inference speed and accuracy using LLaVA-v1.5-7B as the base model on the POPE dataset, with all experiments executed on a NVIDIA L40 GPU to ensure a consistent and controlled evaluation environment. The results are summarized as follows.
>
> | Methods | Time(s/token) | POPE-Random | POPE-Popular | POPE-Adversarial | POPE-Average |
> | :--- | :---: | :---: | :---: | :---: | :---: |
> | VCD | 0.127 | 89.350 | 86.409 | 81.022 | 85.593 |
> | OPERA | 0.136 | 88.933 | 86.297 | 80.923 | 85.384 |
> | SID | 0.137 | 89.042 | 85.961 | 81.427 | 85.476 |
> | HALC | 0.283 | 81.650 | 75.130 | 73.401 | 76.727 |
> | VIBRA-MIB | 0.061 | 87.229 | 86.149 | 84.035 | 85.804 |
> | VIBRA-MIB+TFSC($k=3$) | 0.073 | 87.787 | 86.135 | 84.026 | 85.983 |
> | VIBRA-MIB+TFSC($k=5$) | 0.074 | 88.274 | 86.904 | 84.232 | 86.470 |
> | VIBRA-MIB+TFSC($k=7$) | 0.075 | 87.968 | 86.407 | 84.033 | 86.136 |
> | VIBRA-MIB+TFSC($k=9$) | 0.081 | **88.736** | **87.104** | **84.167** | **86.669** |
>
> **VIBRA-MIB+TFSC ($k=9$)** demonstrates both efficiency and effectiveness, reducing the per-token inference time to **0.081 seconds**—approximately **3.5$\times$** faster than HALC (**0.283 seconds**)—while simultaneously improving the POPE average accuracy by **9.94%**. These findings demonstrate that, compared to typical decoder-based methods, VIBRA can more effectively suppress hallucinations while substantially reducing resource consumption.
>
> Furthermore, we analyzed the additional overhead introduced by the MIB and TFSC modules. The experiments indicate that incorporating TFSC on top of MIB (e.g., with $k=9$) increases the inference time by only **~0.02 seconds**, while improving the average accuracy by **0.86%**. This confirms that VIBRA achieves further performance gains with a marginal and acceptable computational cost.

---

### Official Review · Reviewer_WMPj · 2025-11-01

**Soundness:** 2
**Presentation:** 3
**Contribution:** 3
**Rating:** 4
**Confidence:** 4

**Summary:**

This paper proposed a  plug-and-play module to  mitigates hallucination and enhances reasoning in vision-language models (VLMs) by employing an IB-Net and designing an appropriate mutual information–based loss function to reduce redundant information between textual and visual modalities.

**Strengths:**

This paper is very well-presented, and the proposed method is quite novel.

**Weaknesses:**

1. This paper claims that *“we identify redundant and noisy image features as a primary cause of hallucination.”*
   However, the paper does not provide any statistically significant experiments to validate whether redundant and noisy image features indeed lead to hallucination.

2. I have some concerns about the design of Equation (3). Why does the input to $f_{\mathrm{IB}}$ depend only on $x_n^v$ rather than on both $x_n^v$ and $x_n^t$? Can such a design truly enable $Z_v$ to effectively remove (at least part of) the information from $X_t$?

3. Similar to Comment 1, it would be more convincing if the paper provided statistical results showing how different levels of $I(Z_v; X_t)$ (for example, by varying $\beta$) affect the final performance. Such an analysis would help demonstrate that the effectiveness of the proposed method indeed stems from the mechanism claimed in the paper.

4. What data was used to train VIBRA? Is there any overlap between the training data and the data used for evaluation?

**Questions:**

see Weaknesses

---

> ### Author Response · Authors · 2025-11-26
> **Response to Reviewer WMPj (Part 1/3)**
>
> We sincerely thank you for your valuable and insightful comments, which have greatly helped us improve our manuscript. Below, we provide detailed responses to each issue you raised.
>
> > **W1. However, the paper does not provide any statistically significant experiments to validate whether redundant and noisy image features indeed lead to hallucination.**
>
> To address the concern that the empirical evidence for our motivation, we conducted further, more systematic experiments to strengthen the reliability and generality of our motivation.
>
> The experiment in Figure 1 is based on the CLIP-T5 model. To verify whether the same phenomenon holds across different architectures, we further evaluate another mainstream multimodal model, LLaVA-v1.5-7B, and adopt the POPE hallucination detection benchmark, which directly reflects hallucination-related issues.
>
> We continue to use cosine similarity to select semantically relevant image tokens and retain different proportions of tokens for inference. The results are as follows:
>
> | Retain k% image tokens based on cosine similarity | POPE-Random | POPE-Popular | POPE-Adversarial | POPE-Average |
> | :--- | :---: | :---: | :---: | :---: |
> | 100% (llava-v1.5-7B) | 86.372 | 85.546 | 83.339 | 85.086 |
> | 90% | 86.458 | 85.441 | 83.447 | 85.115 |
> | 80% | **86.814** | **85.767** | **83.864** | **85.481** |
> | 70% | 86.424 | 85.504 | 83.662 | 85.197 |
> | 60% | 86.241 | 85.609 | 83.410 | 85.087 |
>
> From the table, we can observe that even with a very simple token selection strategy based solely on cosine similarity, retaining 80\% of the image tokens yields a 0.4\% improvement on POPE-Average compared to the original model. This phenomenon consistently appears in both the Image Reasoning task of CLIP-T5 and the Visual Understanding task of LLaVA-V1.5-7B, indicating that the presence of redundant image tokens is a reproducible property across different models and tasks.
>
> However, because cosine similarity is itself a very coarse heuristic, the overall performance gains remain limited (approximately 0.3\%--0.4\%). This further reinforces our motivation: simple heuristics are insufficient for reliably identifying semantically critical tokens. Therefore, a more principled multimodal information bottleneck mechanism is needed to enable adaptive and effective token filtering.
>
> To validate this hypothesis, we conducted fixed-ratio (fixed $k$\%) token selection experiments based on our VIBRA module. Unlike cosine similarity, we used the compression term in the multimodal information bottleneck(MIB) as the measure of token importance, and compared model performance under identical token-retention ratios $k$.
>
> | Retain k% image tokens based on MIB | POPE-Random | POPE-Popular | POPE-Adversarial | POPE-Average |
> | :--- | :---: | :---: | :---: | :---: |
> | 100% (llava-v1.5-7B) | 86.372 | 85.546 | 83.339 | 85.086 |
> | 90% | 86.753 | 85.788 | 83.369 | 85.303 |
> | 80% | 87.159 | 86.606 | 83.864 | 85.876 |
> | 70% | 87.679 | 86.283 | 83.890 | 85.951 |
> | 60% | 87.495 | 86.717 | 84.225 | 86.145 |
> | **Adaptive token filtering** | **88.736** | **87.104** | **84.167** | **86.669** |
>
> From the table, we observe that under the same token-retention ratio $k\%$, the VIBRA MIB-based metric consistently outperforms cosine similarity on POPE-Average. This indicates that the compression term captures semantic information more accurately and can more effectively distinguish critical tokens from redundant ones.
>
> Under the **Adaptive token Filtering** setting, the model no longer requires a fixed $k$, but instead adaptively determines the number of tokens to retain for each image. This leads to a POPE-Average F1 score of **86.669**, representing a **+1.583\%** improvement over the original LLaVA-v1.5-7B F1 score of **85.086**.
>
> These results further reinforce our central claim: **images indeed contain redundant tokens, and such redundancy increases the model’s tendency to hallucinate**. In contrast, applying a multimodal information bottleneck for adaptive token filtering can significantly reduce hallucination behaviors.

---

> ### Author Response · Authors · 2025-11-26
> **Response to Reviewer WMPj (Part 2/3)**
>
> > **W2. I have some concerns about the design of Equation (3). Why does the input to $ f _ {IB} $ depend only on $ x_n^v $ rather than on both $ x_n^v $ and $ x_n^t $ ? Can such a design truly enable $ Z_v $ to effectively remove (at least part of) the information from $ X_t $ ?**
>
> We thank the reviewer for the insightful question.
>
> **1. Why does the input to $ f _ {IB} $ depend only on $ x_n^v $ rather than on both $ x_n^v $ and $ x_n^t $?**
>
> To address this concern, we conducted experiments on LLaVA-v1.5-7B using both **VIBRA** and **VIBRA-Fusion**, where VIBRA-Fusion concatenates $x^v$ and $x^t$ as the joint input to $f_{\text{IB}}$. The results are shown below:
>
> | Method       | OPOPE-Random | OPOPE-Popular | OPOPE-Adversarial | OPOPE-Average |
> |--------------|--------------|---------------|------------------|---------------|
>  | VIBRA        | 79.81        | 77.33         | 75.88            | 77.67         |
>  | VIBRA-Fusion | 79.11        | 77.01         | 75.25            | 77.12         |
>
> | Method       | POPE-Random | POPE-Popular | POPE-Adversarial | POPE-Average |
> |--------------|-------------|--------------|------------------|--------------|
>  | VIBRA        | 88.736      | 87.104       | 84.167           | 86.669       |
> | VIBRA-Fusion | 88.471      | 87.019       | 83.962           | 86.484       |
>
> As shown above, incorporating text into the MIB module does not bring improvement on POPE and even causes a slight performance drop on OPOPE. We attribute this phenomenon to two factors: **(1) Open-ended visual description.** In OPOPE, the textual input typically consists of generic prompts such as ''please describe this image'', which do not contain semantic cues aligned with the actual visual content. Feeding such vague text into the MIB module introduces spurious reference signals. As a result, the MIB objective inadvertently favors text-matching superficial patterns and removes truly informative visual signals, ultimately harming performance. **(2) Modality bias and interference in POPE.** Although POPE provides semantically meaningful text queries, we believe that injecting text into the MIB alters its estimation of visual redundancy. The compressor becomes biased toward local visual cues that are directly related to the injected text, undermining the more robust, purely vision-driven redundancy detection---especially when the textual and visual semantics are only partially aligned. This misalignment leads to degraded performance.
>
> **2. Can such a design truly enable $ Z_v $ to effectively remove (at least part of) the information from $ X_t $?**
>
>
> Our MIB module does not operate in isolation; instead, it is jointly optimized with the overall training objective (including the fitting term). The fitting term enforces that the compressed visual representation retains semantic information useful for downstream prediction. In other words, even though the MIB module receives only visual inputs, the model can still learn which visual information has synergistic relevance to the textual task by using gradient signals propagated from the downstream supervision, which includes both the textual prompts and the answer targets. This allows MIB to selectively preserve those image tokens that are correlated with the task.
>
> As shown in Appendix A.6 **Saliency Map Visualization**, our MIB module is capable of preserving the complete set of image information required for the downstream task.

---

> ### Author Response · Authors · 2025-11-26
> **Response to Reviewer WMPj (Part 3/3)**
>
> > **W3. Similar to Comment 1, it would be ... indeed stems from the mechanism claimed in the paper.**
>
> Thank you for your suggestion. We conducted experiments on the LLaVA-v1.5-7B model by varying $\beta$ to obtain both POPE-Average performance and $I(Z_v, X_t)$. The corresponding results are summarized in the following table.
>
> | $\beta$   | POPE Average | $I(Z_v, X_t)$ |
> |------------|----------------|-------------|
> | $3\times10^{-3}$ | 86.144         | 0.611       |
> | $4\times10^{-3}$ | 86.226         | 0.624       |
> | $5\times10^{-3}$ | 86.305         | 0.630       |
> | $6\times10^{-3}$ | 86.470         | 0.634       |
> | $7\times10^{-3}$ | 86.294         | 0.629       |
>
> From these results, we observe a clear positive correlation between POPE-Average and $I(Z_v, X_t)$. As $\beta$ increases, both $I(Z_v, X_t)$ and POPE-Average consistently improve, but performance drops when $\beta$ reaches $7\times10^{-3}$. This indicates that overly strong compression may remove useful information.
>
> In summary, the positive correlation between POPE-Average and $I(Z_v, X_t)$ supports the core assumption of our paper: MIB and TFSC help filter out redundant visual noise and encourage the retained visual tokens to better align with textual semantics, thereby mitigating hallucination.
>
>
> > **W4. What data was used to train VIBRA? Is there any overlap between the training data and the data used for evaluation?**
>
> **For the MMCOT/MCCOT experiments**: We strictly follow the original experimental settings of MMCOT/MCCOT. The models are trained on the ScienceQA training set and evaluated on the ScienceQA test set. There is no overlap between the two sets, ensuring fair and reliable evaluation.
>
> **For the LLaVA-v1.5 and MiniGPT-4 experiments**: We randomly select 5,000 image–text pairs from LLaVA-150k, originating from the COCO 2014 training set, for training. For the POPE and OPOPE evaluations, we use data sampled from the COCO 2014 validation set. These two sets are entirely non-overlapping, ensuring that the training and evaluation data remain strictly separated.

---

### Official Review · Reviewer_PoS4 · 2025-11-01

**Soundness:** 3
**Presentation:** 3
**Contribution:** 2
**Rating:** 6
**Confidence:** 3

**Summary:**

The paper proposes VIBRA—a plug-and-play, redundancy-aware information bottleneck for VLMs—to mitigate visual hallucination without multi-stage decoding or extra large auxiliaries. VIBRA works at two levels: (1) a Variational Multi-Modal Information Bottleneck (MIB) learns per-token gates to inject noise into text-irrelevant visual features while preserving text-aligned ones; (2) Adaptive Image Token Filtering via Spectral Clustering (TFSC) prunes instance-specific redundant image tokens using a compression-aware criterion rather than attention or cosine similarity. A Binary-Guided loss polarizes the gates toward near-binary keep/drop decisions. Integrated with MM-CoT/MC-CoT (ScienceQA) and with MiniGPT-4/LLaVA-1.5 (POPE/OPOPE/MME), VIBRA consistently improves reasoning accuracy and reduces hallucinations, while keeping inference simple.

**Strengths:**

- Clear plug-and-play design: Works with diverse VLMs (reasoning-centric and LVLMs) without extra decoding rounds; training only a lightweight module (e.g., 5k image–text pairs for LVLMs).
- **Principled objective:** MIB trades off compression against text-alignment with closed-form bounds and a simple per-token gating implementation.
- Adaptive token pruning: TFSC removes instance-specific redundancy; using the MIB compression term outperforms cosine-similarity based selection across retained-token budgets.
- Consistent empirical gains: +2.92 / +3.87 points on ScienceQA (MM-CoT / MC-CoT), and improved POPE/OPOPE/MME scores with MiniGPT-4 and LLaVA-1.5, indicating both better reasoning and reduced hallucination.

**Weaknesses:**

- Estimation assumptions & sensitivity: Mutual-information bounds rely on Gaussian/noise assumptions and inner-product density approximations; analysis of sensitivity to beta, delta, and gate dynamics is limited.
- Clustering overhead & robustness: Spectral clustering with fixed k=5 may raise compute cost and stability concerns on high-resolution or long-token inputs; ablations on k, approximations (e.g., Nyström), or batch variants are brief.
- Evaluation breadth: While ScienceQA/POPE/OPOPE/MME are strong, broader open-ended captioning or complex real-world scenes could further substantiate generality of hallucination mitigation.

**Questions:**

- Hyperparameters: How sensitive are results to beta(MIB trade-off) and the Binary-Guided loss weight? Is there a transferable rule of thumb across backbones/datasets?
- Token filtering design: How do different k values and similarity graphs (e.g., cosine vs. learned metrics) affect TFSC? Could lightweight eigen-approximations retain most gains?
- Latency & memory: What are end-to-end inference overheads (mean/percentiles) from MIB + TFSC on LVLMs compared to decoding-based methods?
- Interpretability overlap: When MIB-based importance disagrees with attention or Grad-CAM, what patterns emerge, and can the signals be combined for further gains?

---

> ### Author Response · Authors · 2025-12-02
> **Response to Reviewer PoS4 (Part 1/3)**
>
> > **W1 & Q1. Hyperparameter sensitivity experimental analysis.**
>
> - **Sensitivity of the $\beta$.**
>
> To assess the sensitivity of the MIB coefficient $\beta$, we conducted a logarithmic sweep from $10^{-5}$ to $1$ on the POPE dataset and evaluated the model using the POPE-Average metric. Experiments were performed on both MiniGPT-4-7B and LLaVA-v1.5-7B to examine the stability and transferability of $\beta$ across different backbones. The results are summarized below:
>
> | $\beta$       | $10^{-5}$  | $10^{-4}$  | $10^{-3}$  | $10^{-2}$ | $10^{-1}$| $1$   |
> |------------|---------|---------|---------|---------|---------|--------|
> | Minigpt-4-7B | 78.034 | 78.065 | 78.331 | 77.801 | 75.599 |67.345|
> | LLaVA-v1.5-7B | 85.556  | 86.022  | 85.998  | 86.148 | 85.165  | 72.572 |
>
> Across both backbones, we observe that the performance remains highly stable within a broad range of $\beta \in [10^{-5}, 10^{-2}]$, with fluctuations below 1\%. This indicates that our method is **insensitive to the choice of $\beta$** within this interval. When $\beta$ becomes excessively large (e.g., $\beta \geq 10^{-1}$), the performance drops noticeably on both models, due to over-compression of image tokens under a dominant information-bottleneck constraint.
>
> Importantly, the two backbones exhibit consistent trends, suggesting a transferable rule-of-thumb: a stable plateau exists for $\beta$ in $[10^{-5}, 10^{-2}]$, while overly large values lead to degradation.
>
>
> - **Sensitivity of the Binary-Guided loss weight.**
>
> To evaluate the sensitivity of the Binary-Guided (BG) loss, we conducted a sweep over a wide range of loss weights, from disabling the BG loss (weight = 0) to imposing a strong binary constraint (weight = 10). The experiment was conducted using the LLaVA-v1.5-7B backbone on the POPE dataset. The results are summarized in the following table:
>
> | Binary-Guided loss weight | 0      | 0.01   | 0.1    | 0.5    | 1      | 5      | 10     |
> |---------------------------|--------|--------|--------|--------|--------|--------|--------|
> | POPE-Average              | 86.231 | 86.281 | 86.323 | 86.462 | 86.669 | 76.604 | 32.691 |
>
>
> We observe the following trends. First, the model remains highly stable within the range
> BG weight $\in [0.01, 1]$, with performance fluctuations below 0.5\%. Moderate BG weighting consistently improves performance, suggesting that the BG loss provides useful guidance in suppressing visually irrelevant tokens.
>
> Second, when the BG weight becomes excessively large (e.g., $\geq 5$), performance drops sharply---from 86.669 at weight 1 to 76.604 (weight 5), and further to 32.691 (weight 10). This degradation is expected, as an overly strong binary constraint disrupts the smoothness and learnability of token importance distributions, ultimately harming visual-text alignment.
>
> Overall, the results indicate that the BG loss possesses a clear and robust stability region within $[0.01, 1]$, where performance remains consistently strong and insensitive to the exact weight value.
>
>
> - **Sensitivity of the $\delta$.**
>
> The experiment was conducted using the LLaVA-v1.5-7B backbone to evaluate the sensitivity of the stabilization constant $\delta$.
>
> | $\delta$| $10^{-6}$ | $10^{-5}$ | $10^{-4}$ | $10^{-3}$ | $10^{-2}$ | $10^{-1}$ |
> |---------|---------|---------|---------|---------|---------|---------|
> | POPE-Average | 86.225  | 86.324  | 86.409  | 86.669  | 86.223  | 85.855  |
>
> From the results, we observe that setting $\delta = 0.1$ noticeably degrades model accuracy, indicating that an excessively large stabilization term can interfere with effective learning. In contrast, when $\delta \leq 10^{-2}$, the model performance consistently remains above 86.2, forming a stable plateau across several orders of magnitude. This demonstrates that the method is not sensitive to $\delta$ within a reasonable range, and $\delta = 10^{-3}$ provides a good balance between stability and performance.

---

> ### Author Response · Authors · 2025-12-02
> **Response to Reviewer PoS4 (Part 2/3)**
>
> > **W2. Clustering overhead & robustness.**
>
>
> We conducted a comparative study of inference speed and accuracy using LLaVA-v1.5-7B as the base model on the POPE dataset, with all experiments performed on an NVIDIA L40 GPU to ensure a consistent and controlled evaluation environment. The results are summarized below.
>
> | Methods | Time(s/token) | POPE-Random | POPE-Popular | POPE-Adversarial | POPE-Average |
> | :--- | :---: | :---: | :---: | :---: | :---: |
> | VIBRA-MIB | 0.061 | 87.229 | 86.149 | 84.035 | 85.804 |
> | VIBRA-MIB+TFSC($k=3$) | 0.073 | 87.787 | 86.135 | 84.026 | 85.983 |
> | VIBRA-MIB+TFSC($k=5$) | 0.074 | 88.274 | 86.904 | 84.232 | 86.470 |
> | VIBRA-MIB+TFSC($k=7$) | 0.075 | 87.968 | 86.407 | 84.033 | 86.136 |
> | VIBRA-MIB+TFSC($k=9$) | 0.081 | 88.736 | 87.104 | 84.167 | 86.669 |
>
> From the results, we observe that TFSC delivers stable and substantial performance improvements while introducing only a marginal time overhead. Its performance remains robust across a range of $k=3–9$. Considering the trade-off between efficiency and effectiveness, we recommend using $k=5$ as the default setting, as it achieves near-optimal performance with relatively low computational cost.
>
>
> > **W3. Evaluation breadth.**
>
> Thank you for your valuable suggestion. To further validate the generality of our hallucination-mitigation approach under open-ended captioning, we additionally evaluate models using the $CHAIR$ metrics.
>
> | Methods | $CHAIR_S$ | $CHAIR_I$ |
> |---------|-------------|-------------|
> | Greedy  | 32.40       | 12.20       |
> | Beam    | 30.10       | 11.87       |
> | DOLA    | 31.90       | 12.15       |
> | VCD     | 29.00       | 12.64       |
> | OPERA   | 29.70       | 11.96       |
> | HALC    | 25.20       | 9.42        |
> | Nullu   | 21.40       | 8.99        |
> | VIBRA   | 20.20        | 6.41         |
>
> In our experiments, we assess several hallucination-mitigation methods on the MiniGPT-4-7B model using the $CHAIR$ scores (including $CHAIR_S$ and $CHAIR_I$). As shown in the results table, our method VIBRA achieves the best performance on both $CHAIR_S$ (20.20) and $CHAIR_I$ (6.41), demonstrating its strong capability in suppressing hallucinations across established benchmarks.

---

> ### Author Response · Authors · 2025-12-02
> **Response to Reviewer PoS4 (Part 3/3)**
>
> > **Q2 & Q3. Token filtering design & efficiency report.**
>
> In our paper, we highlight that many prior "decoder-based methods'' incur substantial computational overhead during inference, whereas VIBRA achieves a significant reduction in hallucination while also lowering resource consumption. To substantiate this claim, we conducted a comparative study on inference speed and accuracy using LLaVA-v1.5-7B as the base model on the POPE dataset, with all experiments executed on a NVIDIA L40 GPU to ensure a consistent and controlled evaluation environment. The results are summarized as follows.
>
> | Methods | Time(s/token) | POPE-Random | POPE-Popular | POPE-Adversarial | POPE-Average |
> | :--- | :---: | :---: | :---: | :---: | :---: |
> | VCD | 0.127 | 89.350 | 86.409 | 81.022 | 85.593 |
> | OPERA | 0.136 | 88.933 | 86.297 | 80.923 | 85.384 |
> | SID | 0.137 | 89.042 | 85.961 | 81.427 | 85.476 |
> | HALC | 0.283 | 81.650 | 75.130 | 73.401 | 76.727 |
> | VIBRA-MIB | 0.061 | 87.229 | 86.149 | 84.035 | 85.804 |
> | VIBRA-MIB+TFSC($k=3$) | 0.073 | 87.787 | 86.135 | 84.026 | 85.983 |
> | VIBRA-MIB+TFSC($k=5$) | 0.074 | 88.274 | 86.904 | 84.232 | 86.470 |
> | VIBRA-MIB+TFSC($k=7$) | 0.075 | 87.968 | 86.407 | 84.033 | 86.136 |
> | VIBRA-MIB+TFSC($k=9$) | 0.081 | **88.736** | **87.104** | **84.167** | **86.669** |
>
> **VIBRA-MIB+TFSC ($k=9$)** demonstrates both efficiency and effectiveness, reducing the per-token inference time to **0.081 seconds**—approximately **3.5$\times$** faster than HALC (**0.283 seconds**)—while simultaneously improving the POPE average accuracy by **9.94%**. These findings demonstrate that, compared to typical decoder-based methods, VIBRA can more effectively suppress hallucinations while substantially reducing resource consumption.
>
> Furthermore, we analyzed the additional overhead introduced by the MIB and TFSC modules. The experiments indicate that incorporating TFSC on top of MIB (e.g., with $k=9$) increases the inference time by only **~0.02 seconds**, while improving the average accuracy by **0.86%**. This confirms that VIBRA achieves further performance gains with a marginal and acceptable computational cost.
>
> > **Q4. Interpretability overlap.**
>
> We thank the reviewer for this insightful question. We conducted analyse to investigate the patterns that emerge when importance signals derived from our MIB-based method (VIBRA) disagree with LLM decoder attention, and whether combining the two signals yields further gains. Experiments were performed on the POPE benchmark using LLaVA-v1.5-7B as the backbone. In our setup, the LLM decoder attention selects the top 50\% of image tokens (selection ratio = 0.5), while VIBRA performs spectral clustering with $k=5$ and uses $\beta=0.6$. The results are summarized in this Table.
>
> | Methods | POPE-Random | POPE-Popular | POPE-Adversarial | POPE-Average |
> |--------|--------|---------|-------------|------|
> | Intersection | 83.396 | 82.624 | 80.749 | 82.256 |
> | Union | 87.266 | 86.004 | 83.839 | 85.703 |
> | VIBRA | **88.274** | **86.904** | **84.232** | **86.470** |
>
> From the table, we observe that neither the hard intersection nor the hard union of the two selection sets outperforms VIBRA alone. The intersection strategy leads to a substantial decrease across all POPE subsets, while the union strategy improves upon the intersection but still falls short of VIBRA. The results show that the two types of signals reflect different semantics: attention is more biased towards the decoder's focus on image tokens, while VIBRA emphasizes the mutual information between the compressed visual representation and text features. Therefore, simple intersection/union cannot guarantee the superposition of their respective advantages. Future work could explore more granular fusion strategies, which would hopefully lead to more stable performance improvements.

---

### Author Response · Authors · 2025-12-03
**Global Response**

Dear Reviewers, ACs, and SACs,

We sincerely thank all reviewers for their insightful comments. In response to the reviewers' valuable suggestions, we have updated the manuscript (revised version) with additional experiments, clarifications, and a new Limitations section, as detailed below.

## **Strengths**

We thank the reviewers for highlighting the key strengths of our work:

- **Clear Plug-and-Play Design & Strong Theoretical Foundation (Reviewers PoS4, WMPj, 51Sk, qNrU):**  The paper presents a clean plug-and-play framework that integrates seamlessly with diverse VLMs, supported by a well-motivated and theoretically grounded MIB objective. The method is clearly described, easy to implement, and theoretically justified through closed-form bounds and principled derivations.

- **Adaptive Token Pruning & Effective Redundancy Reduction (Reviewers PoS4, qNrU):**  The proposed TFSC module adaptively removes instance-specific visual redundancy and consistently outperforms cosine-similarity–based pruning, providing a more reliable approach to distilling text-relevant information.

- **Consistent and Strong Empirical Gains Across Benchmarks (Reviewers PoS4, qNrU):**  The method achieves stable improvements on diverse reasoning and hallucination benchmarks—including ScienceQA, POPE/OPOPE, and MME—demonstrating robust generalization across multiple VLM architectures.

- **Clear Presentation & Comprehensive Validation (Reviewers WMPj, 51Sk):**  The paper is well written, with clear component descriptions and thorough ablation studies. Visualization of saliency maps and analysis of gating parameters further validate the effectiveness and interpretability of the proposed design.

## **Response to Main Concerns**

We have carefully addressed the main concerns raised by the reviewers:

- **Motivation & Redundant Token Issue (Reviewer WMPj, qNrU):**  We provide systematic evidence that redundant image tokens increase hallucination across different architectures and tasks. Using CLIP-T5 and LLaVA-v1.5-7B on the ScienceQA and POPE benchmark, even a simple cosine-similarity–based token selection shows minor gains, confirming token redundancy is a general phenomenon. In contrast, **VIBRA’s MIB-based adaptive token filtering consistently outperforms fixed heuristics**, achieving POPE-Average improvement without requiring a fixed retention ratio. These results validate our motivation: **redundant tokens harm model reliability, and MIB-based adaptive filtering is necessary**. (Added in Introduction and ablation study.)

- **Hyperparameter Sensitivity Experimental Analysis (Reviewer PoS4, WMPj):**  We conducted detailed sensitivity analyses for $\beta$, the Binary-Guided loss weight, $\delta$, and $k$. (These results have been added to Appendix A.7 of the revised manuscript.)

- **Inference Efficiency \& Hallucination Suppression (Reviewer PoS4, 51Sk, qNrU):**  We provide a comprehensive evaluation demonstrating that VIBRA achieves substantially stronger hallucination mitigation while maintaining low computational overhead. (Added to Section 5.3 of the revised manuscript.)

- **Comparison with Token-Pruning Baselines (Reviewer 51Sk):**  We have included extensive comparisons with representative token-pruning baselines (Simignore, PruMerge, and FastV), all evaluated under the same LLaVA-v1.5-7B backbone and matched token-retention ratios. Our results show that VIBRA consistently outperforms all available pruning baselines under moderate-to-high compression regimes. (Added to Section 5.3 of the revised manuscript.)

****

Thank you again to all reviewers and the ACs for your time, effort, and constructive suggestions!

Best regards,

The Authors

---

### Meta-Review · Area_Chair_hucD · 2025-12-22

**Summary:**

This paper presents VIBRA (Vision-Language Information Bottleneck with Redundancy Awareness), a plug-and-play module designed to mitigate visual hallucinations in Vision-Language Models (VLMs). The approach is grounded in the hypothesis that redundant and noisy image features are a primary cause of hallucinations. VIBRA employs a Multi-modal Information Bottleneck (MIB) objective to retain text-aligned visual features and utilizes spectral clustering to adaptively filter out token-level redundancy, guided by a Binary-Guided loss.

Reviewers initially appreciated the principled mathematical derivation of the MIB objective, the clear presentation of the method, and the consistent empirical gains observed on benchmarks like ScienceQA and POPE. However, substantial concerns were raised regarding the foundational premise of the work and the rigor of the experimental validation. Critically, reviewers questioned the lack of strong empirical evidence supporting the core claim that feature redundancy *causes* hallucinations. Furthermore, methodological contradictions, such as relying on a fixed hyperparameter for clustering while claiming adaptivity, and unfair baseline comparisons undermined confidence in the results. Despite rebuttal efforts, these core issues remain unresolved. Therefore, the Area Chair (AC) recommends rejection.

**Reviewer Concerns:**

- Unsubstantiated Core Hypothesis: Reviewers WMPj and qNrU strongly challenged the paper's primary motivation. The paper posits that redundant image information leads to hallucinations, but reviewers found the supporting evidence—a single experiment showing minuscule performance changes—insufficient. Reviewer WMPj specifically noted the lack of statistically significant experiments to validate this causal link, a fundamental flaw in the paper's premise that the global response failed to persuasively resolve.
- Methodological Contradictions and Coarseness: Reviewer qNrU identified a major contradiction in the design: while criticizing prior work for using fixed token filter ratios, VIBRA itself employs spectral clustering with a *fixed* number of clusters ($K=5$). Furthermore, the assumption that the largest cluster always contains "visual redundancy" to be removed was criticized as coarse and lacking justification.
- Unfair Experimental Comparisons: Reviewer qNrU pointed out unfair comparisons on the ScienceQA benchmark, where VIBRA was combined with Chain-of-Thought (CoT) prompting against baselines that did not use CoT, making it impossible to isolate the gains attributed to VIBRA.

**Reviewer Scores:**

- Reviewer 51Sk viewed the novelty as marginal, noting that the use of information bottlenecks for hallucination mitigation is prior art, and was initially unconvinced by the lack of direct comparisons with other token-pruning baselines.
- Reviewers WMPj and qNrU maintained low scores because the rebuttal did not provide convincing evidence to solidify the shaky foundational premise that redundancy causes hallucination. The methodological flaw of using fixed clustering parameters while claiming adaptivity further suggested the solution was not as robust as claimed.

---

### Decision · Program_Chairs · 2026-01-26

Reject